# A Separation in Heavy-Tailed Sampling:
# Gaussian vs. Stable Oracles for Proximal Samplers

**Ye He**
Georgia Institute of Technology
yhe367@gatech.edu

**Alireza Mousavi-Hosseini**
University of Toronto, and Vector Institute
mousavi@cs.toronto.edu

**Krishnakumar Balasubramanian**
University of California, Davis
kbala@ucdavis.edu

**Murat A. Erdogdu**
University of Toronto, and Vector Institute
erdogdu@cs.toronto.edu

## Abstract

We study the complexity of heavy-tailed sampling and present a separation result in terms of obtaining high-accuracy versus low-accuracy guarantees i.e., samplers that require only $\mathcal{O}(\log(1/\varepsilon))$ versus $\Omega(\text{poly}(1/\varepsilon))$ iterations to output a sample which is $\varepsilon$-close to the target in $\chi^2$-divergence. Our results are presented for proximal samplers that are based on Gaussian versus stable oracles. We show that proximal samplers based on the Gaussian oracle have a fundamental barrier in that they necessarily achieve only low-accuracy guarantees when sampling from a class of heavy-tailed targets. In contrast, proximal samplers based on the stable oracle exhibit high-accuracy guarantees, thereby overcoming the aforementioned limitation. We also prove lower bounds for samplers under the stable oracle and show that our upper bounds cannot be fundamentally improved.

## 1 Introduction

The task of sampling from heavy-tailed targets arises in various domains such as Bayesian statistics [GJPS08, GLM18], machine learning [CDV09, BZ17, NŞR19, ŞZTG20, DKTZ20], robust statistics [KN04, JR07, Kam18, YŁR22], multiple comparison procedures [GBH04, GB09], and study of geophysical systems [SP15, QM16, PBEM23]. This problem is particularly challenging when using gradient-based Markov Chain Monte Carlo (MCMC) algorithms due to diminishing gradients, which occurs when the tails of the target density decay at a slow (e.g. polynomial) rate. Indeed, canonical algorithms like Langevin Monte Carlo (LMC) have been empirically observed to perform poorly [LWME19, HMW21, HFBE24] when sampling from such heavy-tailed targets.

Several approaches have been proposed in the literature to overcome these limitations of LMC and related algorithms. The predominant ones include (i) transformation-based approaches, where a diffeomorphic (invertible) transformation is used to first map the heavy-tailed density to a light-tailed one so that a light-tailed sampling algorithm can be used [JG12, YŁR22, HBE24], (ii) discretizing general Itô diffusions with non-standard Brownian motion that have heavy-tailed densities as their equilibrium density [EMS18, LWME19, HFBE24], and (iii) discretizing stable-driven stochastic differential equations [ZZ23]. However, the few theoretical results available on the analysis of algorithms based on approaches (i) and (ii) provide only low-accuracy heavy-tailed samplers; such algorithms require $\text{poly}(1/\varepsilon)$ iterations to obtain a sample that is $\varepsilon$-close to the target in a reasonable metric of choice. Furthermore, quantitative complexity guarantees for the sampling approach used in (iii) are not yet available; thus, existing comparisons are mainly based on empirical studies.

In stark contrast, when the target density is light-tailed it is well-known that algorithms like proximal samplers based on Gaussian oracles and the Metropolis Adjusted Langevin Algorithm (MALA) have high-accuracy guarantees; these algorithms require only $\text{polylog}(1/\varepsilon)$ iterations to obtain a sample which is $\varepsilon$-close to the target in some metric. See, for example, the works by [DCWY19, LST21b,

| | $\nu \geq 1$ | | $\nu \in (0,1)$ | |
|---|---|---|---|---|
| Oracle | Gaussian (Alg. 1) | Stable (Alg. 2 & 3) | Gaussian (Alg. 1) | Stable (Alg. 2 & 3) |
| Complexity | $\tilde{\Omega}(\varepsilon^{-\frac{1}{\nu}})$ (Cor. 2) | $\mathcal{O}(\log(\varepsilon^{-1}))$ (Cor. 5) | $\tilde{\Omega}(\varepsilon^{-\frac{1}{\nu}})$ (Cor. 2) | $\tilde{\mathcal{O}}(\varepsilon^{-\frac{1}{\nu}+1})$ (Cor. 5) |

Table 1: **Separation for Proximal Samplers: Gaussian vs. practical Stable oracles** ($\alpha = 1$)**:** Upper and lower iteration complexity bounds to generate an $\varepsilon$-accurate sample in $\chi^2$-divergence from the generalized Cauchy target densities with degrees of freedom $\nu$, i.e. $\pi_\nu \propto (1 + |x|^2)^{-(d+\nu)/2}$. Here, $\tilde{\Omega}, \tilde{\mathcal{O}}$ hide constants depending on $\nu$ and polylog$(d, 1/\varepsilon)$. For the proximal sampler with a general $\alpha$-Stable oracle (Algorithm 2), the upper bound for $\nu \in (0,1)$ is $\mathcal{O}(\log(1/\varepsilon))$ when $\alpha = \nu$. The lower bounds are from Corollary 2 via $2\mathrm{TV}^2 \leq \chi^2$.

WSC22a, CCSW22, CG23]. Specifically, [LST21b] analyzed the proximal sampling algorithm to sample from a class of strongly log-concave densities and obtained high-accuracy guarantees. [CCSW22] established similar high-accuracy guarantees for the proximal sampler to sample from target densities that satisfy a certain functional inequality, covering a range of light-tailed densities with exponentially fast tail decay (e.g. log-Sobolev and Poincaré inequalities). However, it is not clear if the proximal sampler achieves the same desirable performance when the target is not light-tailed.

In light of existing results, in this work, we first consider the following question:

> **Q1.** *What are the fundamental limits of proximal samplers under the Gaussian oracle when sampling from heavy-tailed targets?*

To answer this question, we construct lower bounds showing that Gaussian-based samplers necessarily require poly$(1/\varepsilon)$ iterations to sample from a class of heavy-tailed targets. These results complement the lower bounds on the complexity of sampling from heavy-tailed densities using the LMC algorithm established in [MHFH$^+$23]. With this lower bound in hand, we next consider the following question:

> **Q2.** *Is it possible to design high-accuracy samplers for heavy-tailed targets?*

We answer this in the affirmative by constructing proximal samplers that are based on stable oracles (see Definition 1 and Algorithm 2) by leveraging the fractional heat-flow corresponding to a class of stable-driven SDEs. We analyze the complexity of this algorithm when sampling from heavy-tailed densities that satisfy a fractional Poincaré inequality, and establish that they require only $\log(1/\varepsilon)$ iterations. Together, our answers to **Q1** and **Q2** provide a clear separation between samplers based on Gaussian and stable oracles. Our contributions can be summarized as follows.

- *Lower bounds for the Gaussian oracle*: In Section 2, we focus on **Q1** and establish in Theorems 1 and 2 respectively that the Langevin diffusion and the proximal sampler based on the Gaussian oracle necessarily have a fundamental barrier when sampling from heavy-tailed densities. Our proof technique builds on [Hai10], and provides a novel perspective for obtaining algorithm-dependent lower bounds for sampling, which may be of independent interest.
- *A proximal sampler based on the stable oracle:* In Section 3, we introduce a proximal sampler based on the $\alpha$-stable oracle, which fundamentally relies on the exact implementations of the fractional heat flow that correspond to a stable-driven SDE. Here, the parameter $\alpha$ determines the allowed class of heavy-tailed targets which could be sampled with high-accuracy. In Theorem 3 and Proposition 1, we provide upper bounds on the iteration complexity that are of smaller order than the corresponding lower bounds established for the Gaussian oracle. We provide a rejection-sampling based implementation of the $\alpha$-stable oracle for the case $\alpha = 1$ and prove complexity upper bounds in Corollary 3. Finally, in Theorem 4, considering a sub-class of Cauchy-type targets, we prove lower bounds showing that our upper bounds cannot be fundamentally improved.

An illustration of our results for Cauchy target densities, $\pi_\nu \propto (1 + |x|^2)^{-(d+\nu)/2}$ where $\nu$ is the degrees of freedom, is provided in Table 1. We specifically consider the practical version of the stable proximal sampler with $\alpha = 1$ (i.e., Algorithm 2 with the stable oracle implemented by Algorithm 3), and show that it always outperforms the Gaussian proximal sampler (Algorithm 1). Indeed, when $\nu \geq 1$, the separation between these algorithms is obvious. In the case $\nu \in (0,1)$, Algorithm 2 & 3 has a poly$(1/\varepsilon)$ complexity, nevertheless, it still improves the complexity of the Gaussian proximal sampler by a factor of $\varepsilon$. We also show via lower bounds (in Section 3.4) that the poly$(1/\varepsilon)$ complexity for Algorithm 2 & 3, when $\nu \in (0,1)$, can only be improved up to certain factors. We remark that for the ideal proximal sampler (Algorithm 2), the upper bound when $\nu \in (0,1)$ is also $\mathcal{O}(\log(1/\varepsilon))$. These results demonstrate a clear separation between Gaussian and stable proximal samplers.

**Related works.** We first discuss works analyzing the complexity of heavy-tailed sampling as characterized by a functional inequality assumption. [CDV09] analyzed the connection between sampling

algorithms for a class of $s$-concave densities satisfying a certain isoperimetry condition related to weighted Poincaré inequalities. [HFBE24] undertook a mean-square analysis of discretization of a specific Itô diffusion that characterizes a class of heavy-tailed densities satisfying a weighted Poincaré inequality. [ALPW22] and [ALPW23] analyzed the complexity of pseudo-marginal MCMC algorithms and the random-walk Metropolis algorithm respectively, under weak Poincaré inequalities. As mentioned before, [MHFH+23] showed lower bounds for the LMC algorithm when the target density satisfies a weak Poincaré inequality. [HBE24] and [YŁR22] analyzed a transformation based approach for heavy-tailed sampling under conditions closely related to the same functional inequality. This transformation methodology is also used to demonstrate asymptotic exponential ergodicity for other sampling algorithms like the bouncy particle sampler and the zig-zag sampler, in the heavy-tailed settings [DBCD19, DGM20, BRZ19]. These works provide only low-accuracy guarantees for heavy-tailed sampling and do not consider the use of weak Fractional Poincaré inequalities.

Recent years have witnessed a significant focus on (strongly) log-concave sampling, leading to an extensive body of work that is challenging to encapsulate succinctly. In the context of (strongly) log-concave or light-tailed distributions, a plethora of non-asymptotic investigations have been conducted on LMC variations, including advanced integrators [SL19, LWME19, HBE20], underdamped LMC [CCBJ18, EGZ19, CLW23, DRD20], and MALA [DCWY19, LST20, CLA+21, WSC22b]. Outside the realm of log-concavity, the dissipativity assumption, which regulates the growth of the potential, has been used in numerous studies to derive convergence guarantees [DM17, RRT17, EMS18, EH21, MFWB22, EHZ22, BCE+22].

While research on upper bounds of sampling algorithms' complexity has advanced considerably, the exploration of lower bounds is still nascent. [CGL+22] explored the query complexity of sampling from strongly log-concave distributions in one-dimensional settings. [LZT22] established lower bounds for LMC in sampling from strongly log-concave distributions. [CBL22] presented lower bounds for sampling from strongly log-concave distributions with noisy gradients. [GLL20] focused on lower bounds for estimating normalizing constants of log-concave densities. Contributions by [LST21a] and [WSC22b] provide lower bounds in the metropolized algorithm category, including Langevin and Hamiltonian Monte Carlo, in strongly log-concave contexts. Finally, [CGLL22] contributed to lower bounds in Fisher information for non-log-concave sampling.

## 2 Lower Bounds for Sampling with the Gaussian Oracle

In this section, we focus on **Q1** for both the Langevin diffusion (in continuous time) and the proximal sampler (in discrete time), where both procedures have the target density as their invariant measures. Our results below illustrate the limitation of the Gaussian oracle[1] for heavy-tailed sampling in both continuous and discrete time, showing that the phenomenon is not because of the discretization effect, but is inherently related to the use of Gaussian oracles.

**Langevin diffusion.** We first start with the overdamped Langevin diffusion (LD):

$$\mathrm{d}X_t = -\nabla V(X_t)\mathrm{d}t + \sqrt{2}\mathrm{d}B_t. \tag{LD}$$

LD achieves high-accuracy "sampling" in continuous time, i.e. a $\mathrm{polylog}(1/\varepsilon)$ convergence rate in the light-tailed setting. We make the following dissipativity-type assumption.

**Assumption 1.** *The target density is given by $\pi^X(x) \propto \exp(-V(x))$, where $V : \mathbb{R}^d \to \mathbb{R}$ satisfies*

$$\forall x \in \mathbb{R}^d, \quad \frac{(d+\nu_1)|x|^2}{1+|x|^2} \leq \langle x, \nabla V(x) \rangle \leq \frac{(d+\nu_2)|x|^2}{1+|x|^2} \quad \text{for some } \nu_2 \geq \nu_1 \geq 0.$$

**Remark 1.** *The upper bound on $\langle x, \nabla V(x) \rangle$ ensures that $V$ grows at most logarithmically in $|x|$. Consequently, $\pi^X$ is heavy-tailed and in fact does not satisfy a Poincaré inequality. The lower bound on $\langle x, \nabla V(x) \rangle$ is only needed for deriving the dimension dependency in our guarantees. If one is only interested in the $\varepsilon$ dependency, this condition can be replaced with $0 \leq \langle x, \nabla V(x) \rangle$.*

A classical example of a density satisfying the above assumption is the generalized Cauchy density with degrees of freedom $\nu = \nu_1 = \nu_2 > 0$, where the potential is given by

$$V_\nu(x) := \frac{d+\nu}{2} \ln(1+|x|^2). \tag{1}$$

The following result, proved in Appendix A, provides a lower bound on the performance of LD.

---

[1]Here, for the sake of unified presentation, we refer the use of Brownian motion in (LD) as Gaussian oracle.

---
**Algorithm 1** Gaussian Proximal Sampler [LST21b]

---
**Input:** Sample $x_0$, and step $\eta > 0$. **for** $k = 0, 1, \cdots, N - 1$      // Gibbs sampler

     Sample $y_k \sim \pi^{Y|X}(\cdot|x_k) = \mathcal{N}(x_k, \eta I_d)$             // Heat flow

     Sample $x_{k+1}|y_k \sim \pi^{X|Y}(\cdot|y_k) \propto \pi^X(\cdot) \exp\left(\frac{-|\cdot - y_k|^2}{2\eta}\right)$      // Calls to RGO

**return** $x_N$

---

**Theorem 1.** *Suppose $\pi^X \propto \exp(-V)$ satisfies Assumption 1. Let $X_t$ be the solution of the Langevin diffusion, and $\mu_t := \mathrm{Law}(X_t)$. Then, for any $\delta > 0$,*

$$\mathrm{TV}(\pi^X, \mu_t) \geq C_{\nu_1, \nu_2} d^{\frac{\nu_1 - \nu_2}{2}(1+\delta)} \left(C_\delta(\mu_0) + \kappa_\delta t\right)^{-\frac{\nu_2(1+\delta)}{2}},$$

*where $\kappa_\delta := 1 \vee \frac{2}{d+\nu_2} \vee \frac{\nu_2(1+\delta)}{(d+\nu_2)\delta}$, $C_\delta(\mu_0) := \frac{1}{d+\nu_2}\mathbb{E}[(1+|X_0|^2)^\gamma]^{1/\gamma}$ with $\gamma = \kappa_\delta(d+\nu_2)/2$, and $C_{\nu_1,\nu_2}$ is a constant depending only on $\nu_1$ and $\nu_2$.*

If we assume $|X_0| \leq \mathcal{O}(\sqrt{d})$ for simplicity, then by choosing $\delta = \frac{2 \ln\ln t}{\nu_2 \ln t} \wedge \frac{2 \ln\ln d}{(\nu_2 - \nu_1)\ln d}$, we obtain

$$\mathrm{TV}(\pi^X, \mu_t) \geq \tilde{\Omega}_{\nu_1, \nu_2}(d^{\frac{\nu_1 - \nu_2}{2}} t^{-\frac{\nu_2}{2}}).$$

Thus, LD requires at least $T = \tilde{\Omega}_{\nu_1, \nu_2}\left(d^{\frac{\nu_1 - \nu_2}{\nu_2}}(1/\varepsilon)^{2/\nu_2}\right)$ to reach $\varepsilon$ error in total variation. While this bound may be small in high dimensions when $\nu_2 > \nu_1$, for the canonical model of Cauchy-type potentials with $\nu_2 = \nu_1 = \nu$, it will be independent of dimension, as stated by the following result. Note that Assumption 1 can also cover a general scaling by replacing $|x|$ with $c|x|$ for some constant $c$, which would introduce a multiplicative factor of $1/c^2$ for the lower bound on $T$. This is expected as e.g., mixing to the Gibbs potential $c^2|x|^2$ can be faster than mixing to $|x|^2$ by a factor of $1/c^2$.

**Corollary 1.** *Consider the generalized Cauchy density $\pi_\nu^X \propto \exp(-V_\nu)$ where $V_\nu$ is as in (1). Let $X_t$ be the solution of the Langevin diffusion, and $\mu_t := \mathrm{Law}(X_t)$. For simplicity, assume the initialization satisfies $|X_0| \leq \mathcal{O}(\sqrt{d})$. Then, achieving $\mathrm{TV}(\pi_\nu^X, \mu_T) \leq \varepsilon$ requires $T \geq \tilde{\Omega}_\nu\left(\varepsilon^{-\frac{2}{\nu}}\right)$.*

The above lower bound implies that LD is a low-accuracy "sampler" for this target density in the sense that it depends polynomially on $1/\varepsilon$; this dependence gets worse with smaller $\nu$ as the tails get heavier. It is worth highlighting the gap between the upper bound of [MHFH$^+$23, Corollary 8], which is $\tilde{\mathcal{O}}\left(1/\varepsilon^{4/\nu}\right)$, and the lower bound in Corollary 1.

**Gaussian proximal sampler.** In the remainder of this section, we prove that the Gaussian proximal sampler, described in Algorithm 1, also suffers from a $\mathrm{poly}(1/\varepsilon)$ rate when the target density is heavy-tailed. In each iteration of Algorithm 1, the first step involves sampling a standard Gaussian random variable $y_k$ centered at the current iterate $x_k$ with variance $\eta I$; this is a one-step isotropic Brownian random walk. Alternatively, since the Fokker-Planck equation of the standard Brownian motion is the classical heat equation, this step could also be interpreted as an exact simulation of the heat flow; see, for example, [CG03] and [Wib18]. Specifically, the density of $y_k$ is the solution to the heat flow at time $\eta$ with the initial condition being the density of $x_k$. The second step is called the restricted Gaussian oracle (RGO) as coined by [LST21b]; under which $(x_k, y_k)$ is a reversible Markov chain whose stationary density has $x$-marginal $\pi^X$.

**Assumption 2.** *For some $\nu_2 \geq \nu_1 \geq 0$, the target $\pi^X(x) \propto \exp(-V(x))$ with $V : \mathbb{R}^d \to \mathbb{R}$ satisfies*

$$\forall x \in \mathbb{R}^d \quad \frac{(d+\nu_1)|x|^2}{1+|x|^2} \leq \langle x, \nabla V(x) \rangle, \quad |\nabla V(x)| \leq \frac{(d+\nu_2)|x|}{1+|x|^2}, \quad \Delta V(x) \leq \frac{(d+\nu_2)^2}{1+|x|^2}.$$

The first condition above also appears in Assumption 1 and the second condition implies the upper bound of Assumption 1; thus, the above assumption is stronger. Note that the generalized Cauchy measure (1) satisfies this assumption with $\nu_1 = \nu_2 = \nu$. Under Assumption 2, we state the following lower bound on the Gaussian proximal sampler and defer its proof to Appendix A.

**Theorem 2.** *Suppose $\pi^X \propto \exp(-V)$ satisfies Assumption 2. Let $x_k$ denote the $k^{th}$ iterate of the Gaussian proximal sampler (Algorithm 1) with step $\eta$ and let $\rho_k^X := \mathrm{Law}(x_k)$. Then, for any $\delta > 0$,*

$$\mathrm{TV}(\pi^X, \rho_k^X) \geq C_{\nu_1, \nu_2} d^{\frac{\nu_1 - \nu_2}{2}(1+\delta)} \left(C_\delta(\mu_0) + \kappa_\delta \eta k\right)^{-\frac{\nu_2(1+\delta)}{2}},$$

*where $\kappa_\delta$, $C_\delta(\mu_0)$, and $C_{\nu_1, \nu_2}$ are defined in Theorem 1.*

Above, assuming $|X_0| \leq \mathcal{O}(\sqrt{d})$ with the same choice of $\delta$ as in Theorem 1 yields $\text{TV}(\pi_\nu^X, \rho_k^X) \geq \tilde{\Omega}_{\nu_1,\nu_2}\big(d^{\frac{\nu_1-\nu_2}{2}}(k\eta)^{\frac{-\nu_2}{2}}\big)$. Note that in order for the RGO step to be efficiently implementable, we need to have a sufficiently small $\eta$. The state-of-the-art implementation of RGO requires a step size of order $\eta = \tilde{\mathcal{O}}(1/(Ld^{1/2}))$ when $V$ has $L$-Lipschitz gradients [FYC23]. With this choice of step size, the above lower bound requires at least $N = \tilde{\Omega}_{\nu_1,\nu_2}\big(Ld^{1/2+(\nu_1-\nu_2)/\nu_2}(1/\varepsilon)^{2/\nu_2}\big)$ iterations. The assumptions in Theorem 2 once again cover the canonical examples of generalized Cauchy densities, where we have $L = d + \nu$, which simplifies the lower bound as follows.

**Corollary 2.** *Consider the generalized Cauchy density $\pi_\nu^X \propto \exp(-V_\nu)$ where $V_\nu$ is as in (1). Let $x_k$ denote the $k^{th}$ iterate of the Gaussian proximal sampler, and define $\rho_k^X := \text{Law}(x_k)$, and choose the step size $\eta = \tilde{\mathcal{O}}(1/(Ld^{1/2}))$. If we assume $|X_0| \leq \mathcal{O}(\sqrt{d})$ for simplicity, then achieving $\text{TV}(\pi_\nu^X, \rho_N^X) \leq \varepsilon$ requires $N \geq \tilde{\Omega}_\nu\big(d^{\frac{3}{2}}\varepsilon^{-\frac{2}{\nu}}\big)$ iterations.*

We emphasize that the above lower bound is of order $\text{poly}(1/\varepsilon)$ as advertised. Thus, the RGO-based proximal sampler can only yield a low-accuracy guarantee in this setting.

## 3  Stable Proximal Sampler and the Restricted $\alpha$-Stable Oracle

Having characterized the limitations of Gaussian oracles for heavy-tailed sampling, thereby answering **Q1**, in what follows, we will focus on **Q2** and construct proximal samplers based on the $\alpha$-stable oracle, and prove that they achieve high-accuracy guarantees when sampling from heavy-tailed targets. First, we provide a basic overview of $\alpha$-stable processes and fractional heat flows.

**Isotropic $\alpha$-stable process.** For $t \geq 0$, let $X_t^{(\alpha)}$ be the isotropic stable Lévy process in $\mathbb{R}^d$, starting from $x \in \mathbb{R}^d$, with the index of stability $\alpha \in (0, 2]$, defined uniquely via its characteristic function $\mathbb{E}_x e^{i\langle \xi, X_t^{(\alpha)} - x \rangle} = e^{-t|\xi|^\alpha}$. When $\alpha = 2$, $X_t^{(2)}$ is a scaled Brownian motion, and when $0 < \alpha < 2$, it becomes a pure Lévy jump process in $\mathbb{R}^d$. The transition density of $X_t^{(\alpha)}$ is then given by

$$p^{(\alpha)}(t; x, y) = p_t^{(\alpha)}(y - x) \quad \text{with} \quad p_t^{(\alpha)}(y) = (2\pi)^{-d} \int_{\mathbb{R}^d} \exp(-t|\xi|^\alpha)e^{-i\langle \xi, y \rangle}\mathrm{d}\xi, \quad (2)$$

where the second equation above is the inverse Fourier transform of the characteristic function, thus returns the density. The transition kernel and the density in (2) have closed-form expressions for the special cases $\alpha = 1, 2$. In particular, when $\alpha = 1$, $p_t^{(1)}$ reduces to a Cauchy density with degrees of freedom $\nu = 1$, i.e. $p_t^{(1)}(y) \propto (|y|^2 + t^2)^{-(d+1)/2}$. We finally note that the isotropic stable Lévy process $X_t^{(\alpha)}$ displays self-similarity like the Brownian motion; the processes $X_{at}^{(\alpha)}$ and $a^{1/\alpha}X_t^{(\alpha)}$ have the same distribution. This property is crucial in the development of the stable proximal sampler.

**Fractional heat flow.** The equation $\partial_t u(t, x) = -(-\Delta)^{\alpha/2}u(t, x)$ with the condition $u(0, x) = u_0(x)$ is an extension of the classical heat flow, and is referred to as the fractional heat flow. Here, $-(-\Delta)^{\alpha/2}$ is the fractional Laplacian operator with $\alpha \in (0, 2]$, which is the infinitesimal generator of the isotropic $\alpha$-stable process. For $\alpha = 2$, it reduces to the standard Laplacian operator $\Delta$.

**Stable proximal sampler.** Let $\pi(x, y)$ be a joint density such that $\pi(x, y) \propto \pi^X(x)p^{(\alpha)}(\eta; x, y)$, where $\pi^X$ is the target and $p^{(\alpha)}(\eta; x, y)$ is the transition density of the $\alpha$-stable process, introduced in (2). It is easy to verify that (i) the $X$-marginal of $\pi$ is $\pi^X$, (ii) the conditional density of $Y$ given $X$ is $\pi^{Y|X}(\cdot|x) = p^{(\alpha)}(\eta; x, \cdot)$, (iii) the $Y$-marginal is $\pi^Y = \pi^X * p_\eta^{(\alpha)}$, i.e. $\pi^Y$ is obtained by evolving $\pi^X$ along the $\alpha$-fractional heat flow for time $\eta$, and (iv) the conditional density of $X$ given $Y$ is $\pi^{X|Y}(\cdot|y) \propto \pi^X(\cdot)p^{(\alpha)}(\eta; \cdot, y)$. Based on these, we introduce the following stable oracle.

**Definition 1** (Restricted $\alpha$-Stable Oracle). *Given $y \in \mathbb{R}^d$, an oracle that outputs a random vector distributed according to $\pi^{X|Y}(\cdot|y)$, is called the Restricted $\alpha$-Stable Oracle (R$\alpha$SO).*

Note that when $\alpha = 2$, the R$\alpha$SO reduces to the RGO of [LST21b]. The Stable Proximal Sampler (Algorithm 2) with parameter $\alpha$ is initialized at a point $x_0 \in \mathbb{R}^d$ and performs Gibbs sampling on the joint density $\pi$. In each iteration, the first step involves sampling an isotropic $\alpha$-stable random vector $y_k$ centered at the current iterate $x_k$, which is a one-step isotropic $\alpha$-stable random walk. This could also be interpreted as an exact simulation of the fractional heat flow. Indeed, due to the relation between the fractional heat flow and the isotropic stable process, the density of $y_k$ is exactly the solution to the $\alpha$-fractional heat flow at time $\eta$ with the initial condition being the density of $x_k$.

---
**Algorithm 2** Stable Proximal Sampler with parameter $\alpha$
---
**Input:** Sample $x_0$, step $\eta > 0$, and $\alpha \in (0, 2)$.
**for** $k = 0, 1, \cdots, N - 1$             // Gibbs sampler
     Sample $y_k \sim \pi^{Y|X}(\cdot|x_k) = p^{(\alpha)}(\eta; x_k, \cdot)$      // Fractional heat flow
     Sample $x_{k+1}|y_k \sim \pi^{X|Y}(\cdot|y_k) \propto \pi^X(\cdot)p^{(\alpha)}(\eta; \cdot, y_k)$      // Calls to RαSO
**return** $x_N$

---

When $\alpha = 2$, the first step reduces to an isotropic Brownian random walk and a simulation of the classical heat flow. The second step calls the RαSO at the point $y_k$.

## 3.1 Convergence guarantees

We next provide convergence guarantees for the stable proximal sampler in $\chi^2$-divergence assuming access to the RαSO. Similar results for a practical implementation are presented in Section 3.2. To proceed, we introduce the fractional Poincaré inequality, first introduced in [WW15] to characterize a class of heavy-tailed densities including the canonical Cauchy class.

**Definition 2** (Fractional Poincaré Inequality). *For $\vartheta \in (0, 2)$, a probability density $\mu$ satisfies a $\vartheta$-fractional Poincaré inequality (FPI) if there exists a positive constant $C_{\mathrm{FPI}(\vartheta)}$ such that for any function $\phi : \mathbb{R}^d \to \mathbb{R}$ in the domain of $\mathcal{E}_\mu^{(\vartheta)}$, we have*

$$Var_\mu(\phi) \le C_{\mathrm{FPI}(\vartheta)}\mathcal{E}_\mu^{(\vartheta)}(\phi). \tag{FPI}$$

*where $\mathcal{E}_\mu^{(\vartheta)}$ is a non-local Dirichlet form associated with $\mu$ defined as*

$$\mathcal{E}_\mu^{(\vartheta)}(\phi) := c_{d,\vartheta} \iint_{\{x \ne y\}} \frac{(\phi(x) - \phi(y))^2}{|x - y|^{(d+\vartheta)}} \mathrm{d}x\mu(y)\mathrm{d}y \quad with \quad c_{d,\vartheta} = \frac{2^\vartheta\Gamma((d+\vartheta)/2)}{\pi^{d/2}|\Gamma(-\vartheta/2)|}.$$

**Remark 2.** *FPI is a weaker condition than Assumption 2. In fact, any density satisfying the first 2 conditions in Assumption 2 satisfies $\vartheta$-FPI for all $\vartheta < \nu_1$ [WW15, Theorem 1.1]. In Proposition 2, we show that as $\vartheta \to 2^-$, FPI becomes equivalent to the standard Poincaré inequality.*

In the sequel, $\rho_k^X$ denotes the law of $x_k$, $\rho_k^Y$ denotes the law of $y_k$, and $\rho_k = \rho_k^{X,Y}$ is the joint law of $(x_k, y_k)$. We provide the following convergence guarantee under an FPI, proved in Appendix B.2.

**Theorem 3.** *Assume that $\pi^X$ satisfies the $\alpha$-FPI with parameter $C_{\mathrm{FPI}(\alpha)}$ for $\alpha \in (0, 2)$. For any step size $\eta > 0$ and initial density $\rho_0^X$, the $k^{th}$ iterate of Algorithm 2, with parameter $\alpha$, satisfies*

$$\chi^2(\rho_k^X|\pi^X) \le \exp\left(-k\eta\left(C_{\mathrm{FPI}(\alpha)} + \eta\right)^{-1}\right)\chi^2(\rho_0^X|\pi^X).$$

As a consequence of Remark 2 and Proposition 2, we recover the result in [CCSW22, Theorem 4], by letting $\alpha \to 2^-$. While our results in Theorem 3 are based on Algorithm 2 which requires exact calls to RαSO, the next result, proved in Appendix B.3, shows that even with an inexact implementation of RαSO, the error accumulation is at most linear, and Algorithm 2 still converges quickly.

**Proposition 1.** *Suppose the RαSO in Algorithm 2 is implemented inexactly, i.e. there exists a positive constant $\varepsilon_{\mathrm{TV}}$ such that $\mathrm{TV}(\tilde{\rho}_k^{X|Y}(\cdot|y), \rho_k^{X|Y}(\cdot|y)) \le \varepsilon_{\mathrm{TV}}$ for all $y \in \mathbb{R}^d$ and $k \ge 1$, where $\tilde{\rho}_k^{X|Y}(\cdot|y)$ is the density of the inexact RαSO sample conditioned on $y$. Let $\tilde{\rho}_k^X$ be the density of the output of the $k^{th}$ step of Algorithm 2 with the inexact RαSO and $\rho_k^X$ be the density of the output of $k^{th}$ step Algorithm 2 with the exact RαSO. Then, for all $k \ge 0$,*

$$\mathrm{TV}(\tilde{\rho}_k^X, \rho_k^X) \le \mathrm{TV}(\tilde{\rho}_0^X, \rho_0^X) + k\,\varepsilon_{\mathrm{TV}}.$$

*Further, if $\tilde{\rho}_0^X = \rho_0^X$, for any $K \ge K_0$, we get $\mathrm{TV}(\tilde{\rho}_X^K, \pi^X) \le \varepsilon$, if $\varepsilon_{\mathrm{TV}} \le \varepsilon/2K$, where the constant $K_0 = (1 + C_{\mathrm{FPI}(\alpha)}\eta^{-1})\log\left(\chi^2(\tilde{\rho}_0^X|\pi^X)/\varepsilon^2\right)$ with $C_{\mathrm{FPI}(\alpha)}$ being the $\alpha$-FPI parameter of $\pi^X$.*

## 3.2 A practical implementation of RαSO

In the sequel, we introduce a practical implementation of RαSO when $\alpha = 1$. For this, we consider the case when the target density $\pi^X \propto e^{-V}$ satisfies the 1-FPI with parameter $C_{\mathrm{FPI}(1)}$. A more thorough implementation of RαSO for other values of $\alpha$ will be investigated in future work.

**Assumption 3.** *There exist constants $\beta, L > 0$ such that for any minimizer $x^* \in \arg\min_{y \in \mathbb{R}^d} V(y)$ and for all $x \in \mathbb{R}^d$, $V$ satisfies $V(x) - V(x^*) \le L|x - x^*|^\beta$.*

---

**Algorithm 3** R$\alpha$SO Implementation for $\alpha = 1$ via Rejection Sampling

---

**Input:** $V, x^* \in \arg\min V, \eta > 0, y \in \mathbb{R}^d$.
**while** TRUE                                              *// Rejection sampling*
    Generate $(Z_1, Z_2, u) \sim \mathcal{N}(0, I_d) \otimes \mathcal{N}(0, 1) \otimes U[0, 1]$
    $x \leftarrow y + \eta Z_1 / |Z_2|$                    *// Cauchy random vector*
    **return** $x$ **if** $u \leq \exp(-V(x) + V(x^*))$      *// Accept-reject step*

---

Algorithm 3 provides an exact implementation of R$\alpha$SO for $\alpha = 1$ via rejection sampling. Inputs to this algorithm are the intermediate points $y_k$ in the stable proximal sampler (Algorithm 2). Note that Algorithm 3 requires a global minimizer of $V$, which is always assumed to exist, which guarantees that the acceptance probability is non-trivial. It generates proposals with density $p^{(1)}(\eta; \cdot, y)$ and utilizes that $p^{(1)}$ is a Cauchy density and Cauchy random vectors can be generated via ratios between a Gaussian random vector and square-root of a $\chi^2$ random variable. Finally, the accept-reject step ensures that the output $x$ has density $\pi^{X|Y}(\cdot|y) \propto e^{-V} p^{(1)}(\eta; \cdot, y)$. This makes Algorithm 3 a zeroth-order algorithm requiring only access to function evaluations of $V$. Under Assumption 3, by choosing a small step-size, we can control the expected number of rejections in Algorithm 3. We now state the iteration complexity of our stable proximal sampler with this R$\alpha$SO implementation in the following result, whose proof is provided in Appendix B.3.

**Corollary 3.** *Assume $V$ satisfies Assumption 3. If we choose the step-size $\eta = \Theta(d^{-\frac{1}{2}} L^{-\frac{1}{\beta}})$, then Algorithm 3 implements the R$\alpha$SO with $\alpha = 1$, with the expected number of zeroth-order calls to $V$ of order $\mathbb{E}[\exp(L|y_k|^{\beta})]$. Further assume $\pi^X$ satisfies 1-FPI with parameter $C_{\mathrm{FPI}(1)}$. Suppose we run Algorithm 2 with R$\alpha$SO implemented for with $\alpha = 1$ by Algorithm 3. Then, to return a sample which is $\varepsilon$-close in $\chi^2$-divergence to the target, the expected number of iterations required by Algorithm 2 is*

$$\mathcal{O}\big(C_{\mathrm{FPI}(1)} d^{\frac{1}{2}} L^{\frac{1}{\beta}} \log(\chi^2(\rho_0^X | \pi^X)/\varepsilon)\big).$$

Note that the above result provides a high-accuracy guarantee for the implementable version of the stable proximal sampler (Algorithm 3) for a class of heavy-tailed targets, overcoming the fundamental barrier established in Theorem 2 for the Gaussian proximal sampler (i.e., Algorithm 1). A numerical illustration of this improvement is provided in Appendix D by sampling from student-t distributions.

**Remark 3.** *(1) Finding a global minimizer of the potential $V$ can be hard, which could be avoided if a lower bound on the potential $V$ is available; see Appendix B.3. (2) A trivial bound for $\mathbb{E}[\exp(L|y_k|^\beta)]$ is $\exp(LM)$ for $M = \mathbb{E}_{\pi^X}[|X|^\beta] + \chi^2(\rho_0^X|\pi^X)\mathbb{E}_{\pi^X}[|X|^{2\beta}]^{\frac{1}{2}}$. Since our main focus is high vs low accuracy samplers, deriving a sharper bound is beyond the scope of the current paper.*

### 3.3 Illustrative examples

To illustrate our results, we now apply the proximal algorithms to sample from Cauchy densities and discuss the complexity of both the ideal sampler (Algorithm 2) in which we can choose any $\alpha \in (0, 2)$ and the implementable version with $\alpha = 1$ (Algorithm 3). For the ideal sampler, we can choose $\alpha \leq \nu$ for any degrees of freedom $\nu > 0$, and apply Theorem 3 since $\pi_\nu$ satisfies a $\alpha$-FPI [WW15].

**Corollary 4.** *For any $\nu > 0$, consider the generalized Cauchy target $\pi_\nu \propto \exp(-V_\nu)$ with $V_\nu$ defined in (1). For the stable proximal sampler with parameter $\alpha \in (0, 2)$ and $\alpha \leq \nu$ (i.e., Algorithm 2), suppose we set the step-size $\eta \in (0, 1)$ and draw the initial sample from the standard Gaussian density. Then, the number of iterations required by Algorithm 2 to produce an $\varepsilon$-accurate sample in $\chi^2$-divergence is $\mathcal{O}(C_{\mathrm{FPI}(\alpha)}\eta^{-1}\log(d/\varepsilon))$, where $C_{\mathrm{FPI}(\alpha)}$ is the $\alpha$-FPI parameter of $\pi_\nu$.*

For the implementable sampler, since the parameter $\alpha$ is fixed to be 1, whether a suitable FPI is satisfied or not depends on the degrees of freedom $\nu$. Specifically, when $\nu \geq 1$, 1-FPI is satisfied and Corollary 5 applies. When $\nu \in (0, 1)$, on the other hand, 1-FPI is not satisfied. To tackle this issue, we prove convergence guarantees for the proximal sampler under a weak fractional Poincaré inequality; the next corollary, proved in Appendix B.4, summarizes these results.

**Corollary 5.** *For the Cauchy target $\pi_\nu \propto \exp(-V_\nu)$ where $V_\nu$ is defined in (1), we consider Algorithm 2 with $\alpha = 1$, a standard Gaussian initialization, and R$\alpha$SO implemented by Algorithm 3.*

(1) *When $\nu \geq 1$, if we set the step-size $\eta = \Theta\big(d^{-\frac{1}{2}}(d + \nu)^{-4}\big)$, the expected number iterations required by Algorithm 2 to output a sample which is $\varepsilon$-close in $\chi^2$-divergence to the target is of order $\mathcal{O}\big(C_{\mathrm{FPI}(1)} d^{\frac{1}{2}}(d + \nu)^4 \log(d/\varepsilon)\big)$, where $C_{\mathrm{FPI}(1)}$ is the 1-FPI parameter of $\pi_\nu$.*

(2) When $\nu \in (0,1)$, if we set the step-size $\eta = \Theta\big(d^{-\frac{1}{2}}(d+\nu)^{-\frac{4}{\nu}}\big)$, the expected number of iterations required by Algorithm 2, to output a sample which is $\varepsilon$-close in $\chi^2$-divergence to the target is of order $\tilde{\mathcal{O}}\big(\max\big\{c^{\frac{1}{\nu}}d^{\frac{1}{2\nu}+\frac{4}{\nu^2}}, cd^{\frac{1}{2}+\frac{4}{\nu}}\varepsilon^{-\frac{1}{\nu}+1}\big\}\big)$, where $c$ is the positive constant given in (16). Here, $\tilde{\mathcal{O}}$ hides the polylog factors on $d$ and $1/\varepsilon$.

The stable proximal sampler (Algorithm 2) is a high accuracy sampler for the class of generalized Cauchy targets, as long as $\alpha \leq \nu$, meaning that it achieves $\log(1/\varepsilon)$ iteration complexity. The improvement from $\text{poly}(1/\varepsilon)$ to $\log(1/\varepsilon)$ separates the stable proximal sampler and the Gaussian proximal sampler in the task of heavy-tailed sampling. When we use the rejection-sampling implementation with parameter $\alpha = 1$ (Algorithm 3), iteration complexity goes through a phase transition as the tails get heavier. When the generalized Cauchy density has a finite mean ($\nu > 1$), we achieve a high-accuracy sampler with $\log(1/\varepsilon)$ iteration complexity. However, without a finite mean (i.e., $\nu \in (0,1)$), the algorithm becomes a low-accuracy sampler with $\text{poly}(1/\varepsilon)$ complexity. Even in this low-accuracy regime, the implementable stable proximal sampler outperforms the Gaussian one, as originally highlighted in Table 1. Last, we claim that the $\text{poly}(1/\varepsilon)$ complexity of Algorithms 2 and 3 is not due to a loose analysis, as we show $\text{poly}(1/\varepsilon)$ lower bounds in the following section.

### 3.4 Lower bounds for the stable proximal sampler

We now study lower bounds on the stable proximal sampler to sample from the class of target densities satisfying Assumption 2, which includes the generalized Cauchy target. Recall that Assumption 2 implies the FPI used in Theorem 3. The result below, proved in Appendix C, complements Theorem 3, showing the impossibility of achieving $\log(1/\varepsilon)$ rates for a sufficiently large $\alpha$.

**Theorem 4.** *Suppose $\pi^X \propto \exp(-V)$ with $V$ satisfying Assumption 2 and $\frac{\nu_2(d+\nu_2)}{d+\nu_1} < \alpha \leq 2$. Let $x_k$ denote the $k^{th}$ iterate of Algorithm 2 with parameter $\alpha$ and step size $\eta$, and let $\rho_k^X := \text{Law}(x_k)$. Then for any $\tau \in \big(\frac{\nu_2(d+\nu_2)}{d+\nu_1}, \alpha\big)$, and $g(d,\nu_1,\nu_2,\tau) = \nu_2/\{\tau(d+\nu_1) - \nu_2(d+\nu_2)\}$, we have*

$$\text{TV}(\pi^X, \rho_k^X) \geq C_{\nu_1,\nu_2,\alpha}d^{\frac{\tau(d+\nu_1)g(d,\nu_1,\nu_2,\tau)}{2}}\big(\mathbb{E}[(1+|x_0|^2)^{\frac{\tau}{2}}] + m_\tau^{(\alpha)}k^{\frac{\tau}{2}+1}\eta^{\frac{\tau}{\alpha}}\big)^{-(d+\nu_2)g(d,\nu_1,\nu_2,\tau)},$$

*where $C_{\nu_1,\nu_2,\alpha}$ is a constant depending only on $\nu_1,\nu_2,\alpha$, and $m_\tau^{(\alpha)}$ is the $\tau^{th}$ absolute moment of the $\alpha$-stable random variable with density $p_1^{(\alpha)}$ defined in (2).*

**Remark 4.** *The parameter $\tau$ in Theorem 4 can be chosen arbitrarily close to $\alpha$. Specifically, if we assume $|X_0| \leq \mathcal{O}(\sqrt{d})$, then with the choice of $\tau = \alpha - \big(\frac{\log(\log d)}{\log d} \wedge \frac{\log\log(\eta^{-1})}{\log(\eta^{-1})}\big)$, we have*

$$\text{TV}(\pi^X, \rho_k^X) \geq \tilde{\Omega}_{\nu_1,\nu_2,\alpha}\big(d^{\frac{\tau(d+\nu_1)g(d,\nu_1,\nu_2,\alpha)}{2}}\big(d^\alpha + m_\tau^{(\alpha)}k^{\frac{\alpha}{2}+1}\eta\big)^{-(d+\nu_2)g(d,\nu_1,\nu_2,\alpha)}\big),$$

*where $\tilde{\Omega}$ hides polylog$(d/\eta)$ factors.*

The $\tau^{\text{th}}$ absolute moment of the $\alpha$-stable random variable depends on the choice of $\alpha$ and the dimension $d$. It is hard to find an explicit formula of $m_\tau^{(\alpha)}$ in general. An explicit formula is only available in some special cases, such as $\alpha = 1, 2$. Specializing Theorem 4 for the generalized Cauchy potential (i.e., $\nu_1 = \nu_2$) we obtain the following explicit result.

**Corollary 6.** *Let $\alpha \in (0,2]$. Suppose $\pi_\nu \propto \exp(-V_\nu)$ where $V_\nu(x)$ is as in (1) for some $\nu \in (0,\alpha)$. Let $(x_k)_{k \geq 0}$ be the output of Algorithm 2 with parameter $\alpha$ and step-size $\eta > 0$, and $\rho_k^X := \text{Law}(x_k)$ for all $k \geq 0$. Then for any $\tau \in (\nu, \alpha)$,*

$$\text{TV}(\rho_k^X, \pi_\nu) \geq C_{\nu,\alpha}d^{\frac{\nu\tau}{2(\tau-\nu)}}\big(\mathbb{E}[(1+|x_0|^2)^{\frac{\tau}{2}}] + m_\tau^{(\alpha)}k^{\frac{\tau}{2}+1}\eta^{\frac{\tau}{\alpha}}\big)^{-\frac{\nu}{\tau-\nu}}.$$

*where $m_\tau^{(\alpha)}$ is the $\tau^{th}$ absolute moment of the $\alpha$-stable random variable with density $p_1^{(\alpha)}$ as in (2).*

For the rejection sampling implementation in Algorithm 3, $\alpha = 1$ and $m_\tau^{(1)} = \Theta(d^{\frac{\tau}{2}})$ for all $\tau < 1$ (see Appendix B.1). Notice that to implement the R$\alpha$SO in the Stable proximal sampler efficiently, we need a sufficiently small step-size $\eta$. When the target potential satisfies Assumption 3, i.e. $V$ is $\beta$-Hölder continuous with parameter $L$, we require $\eta = \Theta(d^{-\frac{1}{2}}L^{-\frac{1}{\beta}})$ to ensure R$\alpha$SO can be implemented with $\mathcal{O}(1)$ queries. Therefore, if we choose $\eta = \Theta(d^{-\frac{1}{2}}L^{-\frac{1}{\beta}})$, the minimum number of iterations we need to get an $\varepsilon$-error in TV is

$$\Omega_{\nu,\tau}\Big(\varepsilon^{-\frac{2(\tau-\nu)}{(2+\tau)\nu}}d^{\frac{\tau}{2+\tau}}L^{\frac{2\tau}{\beta(2+\tau)}}\Big).$$

For the generalized Cauchy potential with $\nu \in (0,1)$, we have $\beta = \nu/4$ and $L = (d+\nu)/\nu$, which leads to the following corollary.

**Corollary 7.** *Suppose $\pi_\nu^X \propto \exp(-V_\nu)$ is the generalized Cauchy density with $\nu \in (0,1)$. Let $x_k$ denote the $k$-th iterate of the stable proximal sampler with $\alpha = 1$ (Algorithm 3), and $\rho_k^X := \mathrm{Law}(x_k)$. If we choose the step size $\eta = \Theta(L^{-\frac{4}{\nu}} d^{-\frac{1}{2}})$ where $L = \frac{d+\nu}{\nu}$ is the $\nu/4$-Hölder constant of $V_\nu$, and assume, for simplicity, $|x_0| \le \mathcal{O}(\sqrt{d})$, then, $\mathrm{TV}(\pi_\nu^X, \rho_N^X) \le \varepsilon$ requires $N \ge \Omega_{\nu,\tau}\big(d^{\frac{\tau+8\tau/\nu}{2+\tau}} \varepsilon^{-\frac{2(\tau-\nu)}{\nu(2+\tau)}}\big)$, for any $\tau \in (\nu, 1)$. Further, by choosing $\tau = \max(\nu, 1 - \frac{\log(\log(d/\varepsilon))}{\log(d/\varepsilon)})$, we obtain*

$$N \ge \tilde{\Omega}_\nu\Big(d^{\frac{\nu+8}{3\nu}} \varepsilon^{-\frac{2(1-\nu)}{3\nu}}\Big), \quad \text{in order for} \quad \mathrm{TV}(\pi_\nu^X, \rho_N^X) \le \varepsilon.$$

The above result shows that when implementing the R$\alpha$SO in Algorithm 2 with Algorithm 3, to sample from generalized Cauchy targets with $\nu \in (0,1)$, we can at best have an iteration complexity of order $\mathrm{poly}(1/\varepsilon)$, matching the upper bounds in Corollary 5 up to certain factors.

## 4 Overview of Proof Techniques

**Lower bounds.** We build on the techniques developed in [Hai10]. Let $\mu_t$ denotes the law of LD along its trajectory. To proceed, we need some $G : \mathbb{R}^d \to \mathbb{R}$ for which we can upper bound $\mu_t(G) := \int G \mathrm{d}\mu_t$, and some $f : \mathbb{R}^d \to \mathbb{R}$ that satisfies $\pi^X(G \ge y) \ge f(y)$ for all $y \in \mathbb{R}_+$. After finding the candidates $G$ and $f$, Lemma 1 in Appendix A guarantees $\mathrm{TV}(\pi^X, \mu_t) \ge \sup_{y \in \mathbb{R}_+} f(y) - \mu_t(G)/y$. This technique relies on choosing $G$ such that it has heavy tails under $\pi^X$ leading to a large $f(y)$, while having light tails along the trajectory, thus small $\mu_t(G)$. By picking $G = \exp(\kappa V)$ with $\kappa \ge 1$, one can immediately observe that $\pi^X(G) = \infty$, thus $G$ indeed has heavy tails under $\pi^X$.

To control $\mu_t(G)$ along the trajectory, one can use the generator of LD to bound $\partial_t \mu_t(G)$. Recall the generator of LD, $\mathcal{L}_{\mathrm{LD}}(\cdot) = \Delta(\cdot) - \langle \nabla V, \nabla \cdot \rangle$. Therefore, with a choice of $G = \exp(\kappa V)$, controlling $\partial_t \mu_t(G)$ requires bounding the first and second derivatives of $V$. To avoid making extra assumptions for $V$ in the analysis of LD, we instead construct $G$ based on a surrogate potential $\tilde{V}(x) = \frac{d+\nu_2}{2} \ln(1 + |x|^2)$, which is an upper bound to the potential $V$. We then estimate $f$ based on this surrogate potential in Lemma 2, and control the growth of $\mu_t(G)$ in Lemma 3. Combined with Lemma 1, this leads to the proof of Theorem 1, with the details provided in Appendix A.

For the Gaussian proximal sampler, bounding $\rho_k^X(G)$ requires controlling the expectation of $G$ along the forward and backward heat flow. For the particular choice of $G = \exp(\kappa V)$, we show in Lemma 4 that the growth of $\rho_k^X(G)$ can be controlled only by considering a forward heat flow with the corresponding generator $\mathcal{L}_{\mathrm{HF}} = \frac{1}{2}\Delta$. Therefore, given additional estimates on the second derivatives of $V$, we bound the growth of $\rho_k^X(G)$ in Lemma 5. Once this bound is achieved, we can invoke Lemma 1 to finish the proof of Theorem 2.

**Upper bounds.** Our upper bound analysis builds on that by [CCSW22] in the specific ways discussed next. We consider the change in $\chi^2$ divergence when we apply the two operations to the law $\rho_k^X$ to the iterates and the target $\pi^X$: $(i)$ evolving the two densities along the $\alpha$-fractional heat flow for time $\eta$ and $(ii)$ applying the R$\alpha$SO to the resulting densities. For the step $(i)$, it is required to show that the solution along the fractional heat flow of the stable proximal sampler at any time, satisfies FPI. To show this, $(a)$ the convolution property of the FPI is proved in Lemma 6, and $(b)$ the FPI parameter for the stable process follows from [Cha04, Theorem 23]. In Proposition 3, it is then shown that the $\chi^2$ divergence decays exponentially fast along the fractional heat flow under the assumption of FPI. The aforementioned results enable us to prove the exponential decay of $\chi^2$ divergence along the fractional heat flow under FPI in Proposition 3. To deal with the step $(ii)$ above, we use the data processing inequality; see Proposition 3. These two steps together, enable us to derive the stated upper bounds for the stable proximal sampler.

## 5 Discussion

We showed the limitations of Gaussian proximal samplers for high-accuracy heavy-tailed sampling, and proposed and analyzed stable proximal samplers, establishing that they are indeed high-accuracy algorithms. We now list a few important limitations and problems for future research: (i) It is important to develop efficiently implementable versions of the stable proximal sampler for all values of $\alpha \in (0,2)$, and characterize their complexity in terms of problem parameters, (ii) Gaussian

proximal samplers can be interpreted as a proximal point method for approximating the entropic regularized Wasserstein gradient flow of the KL objective [CCSW22]. This leads to the question, *can we provide a variational intepreration of the stable proximal sampler?* A potential approach is to leverage the results by [Erb14] on gradient flow interpretation of jump processes corresponding to the fractional heat equation, (iii) It is possible to use a non-standard Itô process in the proximal sampler (in place of the $\alpha$-stable diffusion); see, for example, [EMS18, LWME19, HFBE24]. With this modification, it is interesting to examine the rates under weighted Poincaré inequalities that also characterize heavy-tailed densities. There are two difficulties to overcome here: $(a)$ How to generate an exact non-standard Itô process? $(b)$ How to implement the corresponding Restricted non-standard Gaussian Oracle, which requires the zeroth order information of the transition density of the Itô process? In certain cases, non-standard Itô diffusion can be interpreted as a Brownian motion on an embedded sub-manifold; thus, the approach in [GLL$^+$23] might be useful.

### Acknowledgements

KB was supported in part by NSF grants DMS-2053918 and DMS-2413426.

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

# A   Lower Bound Proofs for the Langevin Diffusion and the Gaussian Proximal Sampler

While research on upper bounds of sampling algorithms' complexity has advanced considerably, the exploration of lower bounds is still nascent. [CGL$^+$22] explored the query complexity of sampling from strongly log-concave distributions in one-dimensional settings. [LZT22] established lower bounds for LMC in sampling from strongly log-concave distributions. [CBL22] presented lower bounds for sampling from strongly log-concave distributions with noisy gradients. [GLL20] focused on lower bounds for estimating normalizing constants of log-concave densities. Contributions by [LST21a] and [WSC22b] provide lower bounds in the metropolized algorithm category, including Langevin and Hamiltonian Monte Carlo, in strongly log-concave contexts. Finally, [CGLL22] contributed to lower bounds in Fisher information for non-log-concave sampling. In what follows, we take a different approach and rely on the arguments developed in [Hai10].

We begin by stating the following result which drives our lower bound strategy.

**Lemma 1** ([Hai10, Theorem 5.1]). *Suppose $\mu$ and $\nu$ are probability measures on $\mathbb{R}^d$. Consider some $G : \mathbb{R}^d \to \mathbb{R}_+$ and $f : \mathbb{R}_+ \to \mathbb{R}_+$ satisfying $\mu(G \geq y) \geq f(y)$ for all $y \in \mathbb{R}_+$. Then,*

$$\mathrm{TV}(\mu, \nu) \geq \sup_{y \in \mathbb{R}_+} f(y) - \frac{\int G \mathrm{d}\nu}{y}.$$

*In particular, suppose $\mathrm{Id} \cdot f : \mathbb{R}_+ \ni y \mapsto yf(y) \in \mathbb{R}_+$ is a bijection, then*

$$\mathrm{TV}(\mu, \nu) \geq \frac{1}{2} f\Big((\mathrm{Id} \cdot f)^{-1}\big(2m\big)\Big),$$

*for any $m \geq \int G \mathrm{d}\nu$.*

*Proof.* By the definition of total variation and Markov's inequality, for any $y > 0$

$$\mathrm{TV}(\mu, \nu) \geq \mu(G \geq y) - \nu(G \geq y) \geq f(y) - \frac{\int G \mathrm{d}\nu}{y}.$$

When $\mathrm{Id} \cdot f$ is invertible, choosing $y = (\mathrm{Id} \cdot f)^{-1}(2m)$ implies $yf(y) = 2m$ and yields the desired result. $\qquad\square$

To apply Lemma 1 when the target density satisfies Assumption 1, we need to establish tail lower bounds for this density, which we do so via the following lemma. In the following, let $\omega_d := \frac{\pi^{d/2}}{\Gamma((d+2)/2)}$ denote the volume of the unit $d$-ball.

**Lemma 2.** *Suppose $\pi^X(x) \propto \exp(-V(x))$ satisfies Assumption 1. Then, for all $R > 0$,*

$$\pi^X(|x| \geq R) \geq \frac{2de^{-\nu_1/d}}{(d + \nu_1)\Gamma(\nu_1/2)} (d/2)^{\nu_1/2} (1 + R^{-2})^{-(d+\nu_2)/2} R^{-\nu_2}.$$

*When focusing on dependence on $R$ and $d$, we obtain,*

$$\pi^X(|x| \geq R) \geq C_{\nu_1} d^{\nu_1/2} (1 + R^{-2})^{-(d+\nu_2)/2} R^{-\nu_2},$$

*where $C_{\nu_1} = \frac{2^{1-\nu_1/2}e^{-\nu_1}}{(1+\nu_1)\Gamma(\nu_1/2)}$.*

*Proof.* Without loss of generality assume $V(0) = 0$. Via Assumption 1, we have the estimates for $V$,

$$V(x) = \int_{t=0}^{1} \langle x, \nabla V(tx) \rangle \mathrm{d}t \leq (d + \nu_2) \int_{t=0}^{1} \frac{t|x|^2 \mathrm{d}t}{1 + |tx|^2} = \frac{d + \nu_2}{2} \ln(1 + |x|^2),$$

and similarly

$$V(x) \geq \frac{d + \nu_1}{2} \ln(1 + |x|^2).$$

Consequently, using the spherical coordinates,

$$
\begin{aligned}
\pi^X(|x| \geq R) &\geq \frac{1}{Z} \int_{|x| \geq R} (1 + |x|^2)^{-(d+\nu_2)/2} \mathrm{d}x \\
&= \frac{d\omega_d}{Z} \int_{r \geq R} (1 + r^2)^{-(d+\nu_2)/2} r^{d-1} \mathrm{d}r \\
&\geq \frac{d\omega_d (1 + R^{-2})^{-(d+\nu_2)/2}}{Z} \int_{r \geq R} r^{-\nu_2 - 1} \mathrm{d}r \\
&= \frac{d\omega_d (1 + R^{-2})^{-(d+\nu_2)/2}}{Z_d} R^{-\nu_2}.
\end{aligned}
$$

Next, using the lower bound established on $V$ and spherical coordinates, we obtain,

$$
\begin{aligned}
Z &\leq \int_{\mathbb{R}^d} (1 + |x|^2)^{-(d+\nu_1)/2} \mathrm{d}x \\
&= d\omega_d \int_0^\infty (1 + r^2)^{-(d+\nu_1)/2} r^{d-1} \mathrm{d}r \\
&= \frac{1}{2} d\omega_d \int_0^\infty u^{\nu_1/2 - 1} (1 - u)^{d/2 - 1} \mathrm{d}u \\
&= \frac{1}{2} d\omega_d \mathrm{B}(\nu_1/2, d/2) \\
&= \frac{d\omega_d \Gamma(\nu_1/2)\Gamma(d/2)}{2\Gamma((d+\nu_1)/2)},
\end{aligned}
$$

where B denotes the beta function. Plugging back into our tail lower bound, we obtain,

$$
\pi^X(|x| \geq R) \geq \frac{2\Gamma((d+\nu_1)/2)}{\Gamma(\nu_1/2)\Gamma(d/2)} (1 + R^{-2})^{-(d+\nu_2)/2} R^{-\nu_2}.
$$

Moreover, by [MHFH$^+$23, Lemma 32] we have

$$
\frac{\Gamma((d+\nu_1)/2)}{\Gamma(d/2)} = \frac{d}{d+\nu_1} \frac{\Gamma((d+\nu_1+2)/2)}{\Gamma((d+2)/2)} \geq \frac{2de^{-\nu_1/d}}{d+\nu_1} (d/2)^{\nu_1/2},
$$

which completes the proof. $\qquad\square$

Another element of Lemma 1 is controlling the growth of $\mathbb{E}[G(X_t)]$ throughout the process. The following lemma achieves such control under the Langevin diffusion.

**Lemma 3.** *Suppose $(X_t)_{t\geq 0}$ is the solution to the Langevin diffusion starting at $X_0$ with the corresponding potential $V(x)$ satisfying Assumption 1. Let $G(x) = \exp(\kappa \tilde{V}(x))$ where $\tilde{V}(x) = \frac{d+\nu_2}{2} \ln(1 + |x|^2)$ and $\kappa \geq \frac{2}{d+\nu_2} \vee 1$. Then,*

$$
\mathbb{E}[G(X_t)] \leq \left( \mathbb{E}[G(X_0)]^{\frac{2}{\kappa(d+\nu_2)}} + 4\kappa(d+\nu_2)t \right)^{\frac{\kappa(d+\nu_2)}{2}}.
$$

*Proof.* Recall the generator of the Langevin diffusion $\mathcal{L}(\cdot) = \Delta \cdot - \langle \nabla V, \nabla \cdot \rangle$. Then,

$$
\begin{aligned}
\frac{\mathrm{d}\mathbb{E}[G(X_t)]}{\mathrm{d}t} &= \mathbb{E}[\mathcal{L}G(X_t)] \\
&= \kappa \mathbb{E}\left[ \left( \kappa|\nabla\tilde{V}|^2 + \Delta\tilde{V} - \langle \nabla\tilde{V}, \nabla V \rangle \right) G \right] \\
&\leq \kappa \mathbb{E}\left[ \left( \kappa|\nabla\tilde{V}|^2 + \Delta\tilde{V} \right) G \right] \qquad \text{(Assumption 1)} \\
&\leq 2\kappa^2(d+\nu_2)^2 \mathbb{E}\left[ \frac{G(X_t)}{1 + |X_t|^2} \right] \\
&= 2\kappa^2(d+\nu_2)^2 \mathbb{E}\left[ G(X_t)^{1 - \frac{2}{\kappa(d+\nu_2)}} \right] \\
&\leq 2\kappa^2(d+\nu_2)^2 \mathbb{E}[G(X_t)]^{1 - \frac{2}{\kappa(d+\nu_2)}} \qquad \text{(Jensen's Inequality)}.
\end{aligned}
$$

Integrating the above inequality completes the proof. $\qquad\square$

With the above lemmas in hand, we are ready to present the proof of Theorem 1.

*Proof of Theorem 1.* To apply Lemma 1 we choose $G(x) = \exp(\kappa \tilde{V}(x))$ where $\tilde{V}(x) = \frac{d+\nu_2}{2} \ln(1 + |x|^2)$ with $\kappa \geq 1 \vee \frac{2}{d+\nu_2}$. By Lemma 2 we have

$$\pi^X(G(x) \geq y) \geq \pi^X\left(|x| \geq y^{\frac{1}{\kappa(d+\nu_2)}}\right) \geq C_{\nu_1} d^{\nu_1/2}\left(1 + y^{\frac{-2}{\kappa(d+\nu_2)}}\right)^{-(d+\nu_2)/2} y^{\frac{-\nu_2}{\kappa(d+\nu_2)}}.$$

Moreover, define

$$g(t) := \left(g(0)^{\frac{2}{\kappa(d+\nu_2)}} + 4\kappa(d+\nu_2)t\right)^{\frac{\kappa(d+\nu_2)}{2}},$$

with $g(0) := \mathbb{E}[G(X_0)]$. Then by Lemma 3 we have $\mathbb{E}[G(X_t)] \leq g(t)$ and we can invoke Lemma 1 to obtain

$$\mathrm{TV}(\pi^X, \mu_t) \geq \sup_{y \in \mathbb{R}_+} C_{\nu_1} d^{\nu_1/2}\left(1 + y^{\frac{-2}{\kappa(d+\nu_2)}}\right)^{-(d+\nu_2)/2} y^{\frac{-\nu_2}{\kappa(d+\nu_2)}} - \frac{g(t)}{y}.$$

$$\geq \sup_{y \in \mathbb{R}_+} C_{\nu_1} d^{\nu_1/2} \exp\left(-\frac{(d+\nu_2)y^{\frac{-2}{\kappa(d+\nu_2)}}}{2}\right) y^{\frac{-\nu_2}{\kappa(d+\nu_2)}} - \frac{g(t \vee 1)}{y},$$

where we used the fact that $1 + x \leq e^x$ for all $x \in \mathbb{R}$ and $g(t)$ is non-decreasing in $t$. Choose

$$y^* := C'_{\nu_1,\nu_2}\left(\frac{g(t \vee 1)}{d^{\nu_1/2}}\right)^{\frac{\kappa(d+\nu_2)}{\kappa(d+\nu_2)-\nu_2}},$$

for a sufficiently large constant $C'_{\nu_1,\nu_2} \geq 1$. For simplicity, let

$$\tilde{g}(t) := \frac{g(t \vee 1)^{\frac{2}{\kappa(d+\nu_2)}}}{4\kappa(d+\nu_2)},$$

and notice that

$$y^* = C'_{\nu_1,\nu_2} d^{\frac{\kappa(d+\nu_2)}{2} \cdot \frac{\kappa(d+\nu_2)-\nu_1}{\kappa(d+\nu_2)-\nu_2}}\left(4\kappa(1+\nu_2/d)\tilde{g}(t)\right)^{\frac{\kappa^2(d+\nu_2)^2}{2(\kappa(d+\nu_2)-\nu_2)}}. \tag{3}$$

Using the fact that

$$y^* \geq (4\kappa)^{\frac{\kappa^2(d+\nu_2)^2}{2(\kappa(d+\nu_2)-\nu_2)}} d^{\frac{\kappa(d+\nu_2)}{2} \cdot \frac{\kappa(d+\nu_2)-\nu_1}{\kappa(d+\nu_2)-\nu_2}},$$

we have

$$\mathrm{TV}(\pi^X, \mu_t) \geq C_{\nu_1} \exp\left(-\frac{1+\nu_2/d}{8\kappa} \cdot d^{\frac{\nu_1-\nu_2}{\kappa(d+\nu_2)-\nu_2}}\right) d^{\nu_1/2} y^{*\frac{-\nu_2}{\kappa(d+\nu_2)}} - \frac{g(t \vee 1)}{y^*}$$

$$\geq \tilde{C}_{\nu_1,\nu_2} d^{\nu_1/2} y^{*\frac{-\nu_2}{\kappa(d+\nu_2)}} - \frac{g(t \vee 1)}{y^*},$$

where $\tilde{C}_{\nu_1,\nu_2} = C_{\nu_1} e^{-\frac{1+\nu_2/d}{8}}$. By plugging in the value of $y^*$ from (3), we obtain,

$$\mathrm{TV}(\pi^X, \mu_t)$$
$$\geq \left\{\tilde{C}_{\nu_1,\nu_2} C'_{\nu_1,\nu_2}{}^{\frac{-\nu_2}{\kappa(d+\nu_2)}} - C'_{\nu_1,\nu_2}{}^{-1}\right\}\left\{d^{\frac{\nu_1-\nu_2}{2}}\left(2\kappa(1+\nu_2/d)\tilde{g}(t)\right)^{\frac{-\nu_2}{2}}\right\}^{1+\frac{\nu_2}{\kappa(d+\nu_2)-\nu_2}}.$$

Thus for sufficiently large $C'_{\nu_1,\nu_2}$, there exists $C''_{\nu_1,\nu_2}$ such that

$$\mathrm{TV}(\pi^X, \mu_t) \geq C''_{\nu_1,\nu_2}\left\{d^{\frac{\nu_1-\nu_2}{2}}\left(4\kappa(1+\nu/d)\tilde{g}(t)\right)\right\}^{\frac{-\nu_2}{2}}^{1+\frac{\nu_2}{\kappa(d+\nu_2)-\nu_2}}.$$

Choosing $\kappa$ according to the statement of the theorem completes the proof. $\square$

In order to prove a similar theorem for the Gaussian proximal sampler, we control the growth of $\mathbb{E}[G(x_k)]$ for the iterates of the proximal sampler via the following lemmas.

**Lemma 4.** *Suppose $(x_k, y_k)_k$ are the iterates of the Gaussian proximal sampler with step size $\eta$ and target density $\pi^X \propto \exp(-V)$ for some $V : \mathbb{R}^d \to \mathbb{R}$. Let $G(x) = \exp(\kappa V(x))$ with $\kappa \geq 1$. Then, for every $k \geq 0$,*

$$\mathbb{E}[G(x_{k+1})] \leq \mathbb{E}[G(x_k + \sqrt{2\eta}z)],$$

*where $z \sim \mathcal{N}(0, I_d)$ is sampled independently from $x_k$.*

*Proof.* Recall that $\pi^{X|Y}(x|y) \propto \exp\left(-V(x) - \frac{|x-y|^2}{2\eta}\right)$. Therefore,

$$\mathbb{E}[G(x_{k+1}) \mid y_k] = C_{y_k} \int \frac{\exp\left((\kappa - 1)V(x) - \frac{|x-y_k|^2}{2\eta}\right)}{(2\pi\eta)^{d/2}} \mathrm{d}x$$

$$= C_{y_k} \mathbb{E}[G(y_k + \sqrt{\eta}z_1)^{1-1/\kappa} \mid y_k],$$

where $z_1 \sim \mathcal{N}(0, I_d)$. Furthermore,

$$C_{y_k} = \frac{1}{(2\pi\eta)^{d/2}} \int \exp\left(-V(x) - \frac{|x - y_k|^2}{2\eta}\right) \mathrm{d}x$$

$$= \mathbb{E}[G(y_k + \sqrt{\eta}z_1)^{-1/\kappa} \mid y_k].$$

Therefore,

$$\mathbb{E}[G(x_{k+1}) \mid y_k]$$
$$= \frac{\mathbb{E}[G(y_k + \sqrt{\eta}z_1)^{1-1/\kappa} \mid y_k]}{\mathbb{E}[G(y_k + \sqrt{\eta}z_1)^{-1/\kappa} \mid y_k]}$$
$$\leq \mathbb{E}[G(y_k + \sqrt{\eta}z_1) \mid y_k]^{1-1/\kappa} \mathbb{E}[G(y_k + \sqrt{\eta}z_1) \mid y_k]^{1/\kappa} \qquad \text{(Jensen's Inequality)}$$
$$= \mathbb{E}[G(y_k + \sqrt{\eta}z_1) \mid y_k].$$

Recall $y_k = x_k + \sqrt{\eta}z_2$ where $z_2 \sim \mathcal{N}(0, I_d)$ is independent from $x_k$. By the towering property of conditional expectation,

$$\mathbb{E}[G(x_{k+1})] \leq \mathbb{E}[G(x_k + \sqrt{\eta}z_1 + \sqrt{\eta}z_2)]$$
$$= \mathbb{E}[G(x_k + \sqrt{2\eta}z)],$$

where $z \sim \mathcal{N}(0, I_d)$ is independent from $x_k$, which completes the proof. $\qquad\square$

In order to provide a more refined control over $\mathbb{E}[G(x_k)]$, we need additional assumptions on $V$. In particular, when considering the generalized Cauchy density, we arrive at the following lemma.

**Lemma 5.** *Suppose $(x_k, y_k)_k$ are the iterates of the Gaussian proximal sampler with step size $\eta$ and target density $\pi^X \propto \exp(-V)$ satisfies*

$$|\nabla V(x)| \leq \frac{(d + \nu_2)|x|}{1 + |x|^2} \quad \text{and} \quad \Delta V(x) \leq \frac{(d + \nu_2)^2}{1 + |x|^2},$$

*for all $x \in \mathbb{R}^d$. Let $G(x) = \exp(\kappa V(x))$ with $\kappa \geq 1 \vee \frac{2}{d + \nu_2}$. Then, for every $k \geq 0$,*

$$\mathbb{E}[G(x_{k+1})]^{\frac{2}{\kappa(d+\nu_2)}} \leq \mathbb{E}[G(x_k)]^{\frac{2}{\kappa(d+\nu_2)}} + 4\kappa\eta k(d + \nu_2).$$

*Proof.* From Lemma 4, we have

$$\mathbb{E}[G(x_{k+1})] \leq \mathbb{E}[G(x_k + \sqrt{2\eta}z)],$$

where $z \sim \mathcal{N}(0, I_d)$ is independent from $x_k$. Consider the Brownian motion starting at $x_k$, denoted by $Z_t = B_t + x_k$ where $(B_t)$ is a standard Brownian motion in $\mathbb{R}^d$. Notice that the generator for the

process $\mathrm{d}Z_t = \mathrm{d}B_t$ is $\mathcal{L} = \frac{1}{2}\Delta$. Therefore,

$$
\begin{aligned}
\frac{\mathrm{d}\mathbb{E}[G(Z_t)]}{\mathrm{d}t} &= \mathbb{E}[\mathcal{L}G(Z_t)] \\
&= \frac{\kappa}{2}\mathbb{E}\big[G(Z_t)\big(\kappa|\nabla V|^2 + \Delta V\big)\big] \\
&\leq \frac{\kappa(\kappa+1)}{2}\mathbb{E}\Big[G(Z_t)\frac{(d+\nu_2)^2}{1+|Z_t|^2}\Big] \\
&\leq 2\kappa^2(d+\nu_2)^2\mathbb{E}\big[G(Z_t)^{1-\frac{2}{\kappa(d+\nu_2)}}\big] \\
&\leq 2\kappa^2(d+\nu_2)^2\mathbb{E}[G(Z_t)]^{1-\frac{2}{\kappa(d+\nu_2)}} \qquad \text{(Jensen's Inequality)}.
\end{aligned}
$$

Integrating the above inequality yields

$$
\mathbb{E}[G(Z_t)]^{\frac{2}{\kappa(d+\nu_2)}} \leq \mathbb{E}[G(Z_0)]^{\frac{2}{\kappa(d+\nu_2)}} + 2\kappa(d+\nu_2)t.
$$

The proof is complete by noticing that $Z_0 = x_k$ and $Z_t = x_k + \sqrt{2\eta}z$ for $t = 2\eta$. $\qquad\square$

*Proof of Theorem 2.* Notice that the statements of Lemmas 3 and 5 are virtually the same by changing $t$ to $2k\eta$. Using this fact, the rest of the proof follows exactly the same as the proof of Theorem 1. $\quad\square$

## B  Proofs for the Stable Proximal Sampler

### B.1  Preliminaries

In this section, we introduce additional preliminaries on the isotropic $\alpha$-stable process, the fractional Poincaré-type inequalities, the fractional Laplacian and the fractional heat flow.

The Lévy process is a stochastic process that is stochastically continuous with independent and stationary increments. Due to the stochastic continuity, the Lévy processes have càdlàg trajectories, which allows jumps in the paths. A Lévy process $Y_t$ is uniquely determined by a triple $(b, A, \nu)$ through the following Lévy-Khinchine formula: for all $t \geq 0$ and $\xi \in \mathbb{R}^d$,

$$
\mathbb{E}\big[e^{i\langle \xi, Y_t\rangle}\big] = \exp\left(t\big(i\langle b,\xi\rangle - \xi^{\mathsf{T}}A\xi + \int_{\mathbb{R}^d\setminus\{0\}}(e^{i\langle\xi,y\rangle} - 1 - i\langle\xi,y\rangle 1_{\{|y|\leq 1\}}(y))\nu(\mathrm{d}y)\big)\right), \quad (4)
$$

where $b \in \mathbf{R}^d$ is a drift vector. $A \in \mathbb{R}^{d\times d}$ is the covariance matrix of the Brownian motion in the Lévy-Itô decomposition[App09, Thereom 2.4.16] and $\nu$ is the Lévy measure related to the jump parts in the Lévy-Itô decomposition.

The rotationally invariant(isotropic) stable process is a special case for the Lévy process when $b = 0$, $A = 0$ and $\nu$ is the measure given by

$$
\nu(\mathrm{d}y) = c_{d,\alpha}|y|^{-(d+\alpha)}, \quad c_{d,\alpha} = 2^\alpha\Gamma((d+\alpha)/2)/(\pi^{d/2}|\Gamma(-\alpha/2)|). \quad (5)
$$

Based on the Lévy-Khinchine formula (4), if we initialize the process at $x \in \mathbb{R}^d$, its characteristic function is given by

$$
\mathbb{E}_x e^{i\langle\xi, X_t^{(\alpha)} - x\rangle} = e^{-t|\xi|^\alpha}, \qquad x, \xi \in \mathbb{R}^d,\ t \geq 0. \quad (6)
$$

The index of stability $\alpha \in (0, 2]$ determines the tail-heaviness of the densities: the smaller is $\alpha$, the heavier is the tail. The parameter $t$ in (6) measures the spread of $X_t$ around the center. When $\alpha = 2$, the stable process pertains to the Brownian motion running with a time clock twice as fast as the standard one and hence it has continuous paths. When $\alpha \in (0, 2)$, the stable process paths contain discontinuities, which are often referred as jumps. At each fixed time, unlike the Brownian motion, the $\alpha$-stable process density only has a finite $p^{\text{th}}$-moment for $p < \alpha$, i.e.

$$
\mathbb{E}[|X_1^{(\alpha)}|^p] = \begin{cases} +\infty & p \in [\alpha, +\infty), \alpha \in (0, 2), \\ m_p^{(\alpha)} < +\infty & p \in (0, \alpha), \alpha \in (0, 2). \end{cases}
$$

When $d = 1$, the fractional absolute moment formula for $m_p^{(\alpha)}$ can be derived explicitly, see [Nol20, Chapter 3.7]. When $d > 1$, the explicit formula for $m_p^{(\alpha)}$ is only known in some special cases. For example, when $\alpha = 1$, $m_p^{(1)} = \frac{\Gamma((d+p)/2)\Gamma((1-p)/2)}{\Gamma(d/2)\Gamma(1/2)}$ for all $p < 1$. Another good property of $\alpha$-stable process is the self-similarity. By examining the characteristic functions, it is easy to verify that the isotropic $\alpha$-stable process is self-similar with the Hurst index $1/\alpha$, i.e. $X_{at}^{(\alpha)}$ and $a^{1/\alpha}X_t^{(\alpha)}$ have the same distribution. Or equivalently, $p_t^{(\alpha)}(x) = t^{-\frac{d}{\alpha}}p_1^{(\alpha)}(t^{-\frac{1}{\alpha}}x)$ for all $x \in \mathbb{R}^d$ and $t > 0$.

The fractional Laplacian operator in $\mathbb{R}^d$ of order $\alpha$ is denoted by $-(-\Delta)^{\alpha/2}$ for $\alpha \in (0, 2]$. It was introduced as a non-local generalization of the Laplacian operator to model various physical phenomenons. In [Kwa17], ten equivalent definitions of the fractional Laplacian operator are introduced. Here we recall two of them:

(a) Distributional definition: For all Schwartz functions $\phi$ defined on $\mathbb{R}^d$, we have

$$\int_{\mathbb{R}^d} -(-\Delta)^{\alpha/2}f(y)\phi(y)\mathrm{d}y = \int_{\mathbb{R}^d} f(x)\left(-(-\Delta)^{\alpha/2}\phi(x)\right)\mathrm{d}x.$$

(b) Singular integral definition: For a limit in the space $L^p(\mathbb{R}^d)$, $p \in [1, \infty)$, we have

$$-(-\Delta)^{\alpha/2}f(x) = \lim_{r \to 0^+} \frac{2^\alpha \Gamma(\frac{d+\alpha}{2})}{\pi^{d/2}|\Gamma(-\frac{\alpha}{2})|} \int_{\mathbb{R}^d \setminus B_r} \frac{f(x+z) - f(x)}{|z|^{d+\alpha}}\mathrm{d}z.$$

where $B_r$ is the unit ball with radius $r$ centered at the origin.

The fractional Laplacian can be understood as the infinitesimal generator of the stable Lévy process. More explicitly, the semigroup defined by the transition probability $p_t^{(\alpha)}$ in (2) has the infinitesimal generator $-(-\Delta)^{\alpha/2}$, i.e. the density function $p_t^{(\alpha)}$ satisfies the following equation in the sense of distribution, [BHJ08]:

$$\partial_t p_t^{(\alpha)}(x) = -(-\Delta)^{\alpha/2}p_t^{(\alpha)}(x). \tag{7}$$

(7) is usually referred as the $\alpha$-fractional heat flow. When $\alpha = 2$, $-(-\Delta)^{\alpha/2}$ is the Laplacian operator and (7) becomes the heat flow.

**Proposition 2** (From FPI to PI). *When $\vartheta \to 2^-$, the $\vartheta$-FPI reduces to the classical Poincaré inequality with Dirichlet form $\mathcal{E}_\mu(\phi) = \int |\nabla\phi(x)|^2\mathrm{d}x$ for any smooth bounded $\phi : \mathbb{R}^d \to \mathbb{R}^d$.*

*Proof.* It suffices to prove that $\mathcal{E}_\mu^{(\vartheta)}(\phi)$ converges to $\mathcal{E}_\mu(\phi)$ as $\vartheta \to 2^-$ for any smooth function $\phi$. Recall the definition of $\mathcal{E}_\mu^{(\vartheta)}(\phi)$:

$$\mathcal{E}_\mu^{(\vartheta)}(\phi) := c_{d,\vartheta} \iint_{\{x \neq y\}} \frac{(\phi(x) - \phi(y))^2}{|x - y|^{(d+\vartheta)}}\mathrm{d}x\mu(y)\mathrm{d}y \quad \text{with} \quad c_{d,\vartheta} = \frac{2^\vartheta \Gamma((d+\vartheta)/2)}{\pi^{d/2}|\Gamma(-\vartheta/2)|},$$

where $c_{d,\vartheta} = \mathcal{O}(2 - \vartheta)$ as $\vartheta \to 2^-$. Now we rewrite the inside integral in $\mathcal{E}_\mu^{(\vartheta)}(\phi)$ and split the integral region into a centered unit ball, denoted as $B_1$, and its complement:

$$\int_{x \neq y} \frac{(\phi(x) - \phi(y))^2}{|x - y|^{(d+\vartheta)}}\mathrm{d}x = \int_{z \neq 0} \frac{(\phi(y+z) - \phi(y))^2}{|z|^{(d+\vartheta)}}\mathrm{d}z$$

$$= \underbrace{\int_{B_1} \frac{(\phi(y+z) - \phi(y))^2}{|z|^{(d+\vartheta)}}\mathrm{d}z}_{I_1} + \underbrace{\int_{\mathbb{R}^d \setminus B_1} \frac{(\phi(y+z) - \phi(y))^2}{|z|^{(d+\vartheta)}}\mathrm{d}z}_{I_2}.$$

For $I_2$, we have

$$I_2 \leq 4\|\phi\|_\infty^2 \int_{\mathbb{R}^d \setminus B_1} \frac{1}{|z|^{d+\vartheta}}\mathrm{d}z = \frac{4\|\phi\|_\infty^2 d\pi^{\frac{d}{2}}}{\Gamma(\frac{d}{2}+1)} \int_1^\infty r^{-\vartheta+1}\mathrm{d}r = \frac{4\|\phi\|_\infty^2 d\pi^{\frac{d}{2}}}{\vartheta\Gamma(\frac{d}{2}+1)}.$$

As a result, the term in $\mathcal{E}_\mu^{(\vartheta)}(\phi)$ that is induced by $I_2$ satisfies

$$c_{d,\vartheta} \int_{\mathbb{R}^d} I_2\mu(y)\mathrm{d}y \leq c_{d,\vartheta} \frac{4\|\phi\|_\infty^2 d\pi^{\frac{d}{2}}}{\vartheta\Gamma(\frac{d}{2}+1)} \to 0 \quad \text{as } \vartheta \to 2^-.$$

For $I_1$, we have when $\vartheta > 1$,

$$I_1 - \int_{B_1} \frac{|\langle \nabla \phi(y), z \rangle|^2}{|z|^{d+\vartheta}} \mathrm{d}z$$

$$= \int_{B_1} \frac{\big(\phi(y+z) - \phi(y) - \langle \nabla \phi(y), z \rangle\big)\big(\phi(y+z) - \phi(y) + \langle \nabla \phi(y), z \rangle\big)}{|z|^{d+\vartheta}} \mathrm{d}z$$

$$\leq \|\phi\|_{C^2(\mathbb{R}^d)} \|\phi\|_{C^1(\mathbb{R}^d)} \int_{B_1} |z|^{-(d+\vartheta-3)} \mathrm{d}z$$

$$= \|\phi\|_{C^2(\mathbb{R}^d)} \|\phi\|_{C^1(\mathbb{R}^d)} \frac{d\pi^{\frac{d}{2}}}{\Gamma(\frac{d}{2}+1)} \int_0^1 r^{\vartheta-2} \mathrm{d}r$$

$$= \|\phi\|_{C^2(\mathbb{R}^d)} \|\phi\|_{C^1(\mathbb{R}^d)} \frac{d\pi^{\frac{d}{2}}}{(\vartheta-1)\Gamma(\frac{d}{2}+1)},$$

where $\|\phi\|_{C^i(\mathbb{R}^d)} := \sup_{x \in \mathbb{R}^d} |\phi^{(i)}(x)|$ for $i = 1, 2$. As a result, the term in $\mathcal{E}_\mu^{(\vartheta)}(\phi)$ that is induced by $I_1$ satisfies

$$c_{d,\vartheta} \int_{\mathbb{R}^d} \Big(I_2 - \int_{B_1} \frac{|\langle \nabla \phi(y), z \rangle|^2}{|z|^{d+\vartheta}} \mathrm{d}z\Big) \mu(y) \mathrm{d}y \leq c_{d,\vartheta} \frac{\|\phi\|_{C^2(\mathbb{R}^d)} \|\phi\|_{C^1(\mathbb{R}^d)} d\pi^{\frac{d}{2}}}{(\vartheta-1)\Gamma(\frac{d}{2}+1)} \to 0 \quad \text{as } \vartheta \to 2^-.$$

Therefore we have $\mathcal{E}_\mu^{(\vartheta)}(\phi) \to c_{d,\vartheta} \int_{\mathbb{R}^d} \int_{B_1} \frac{|\langle \nabla \phi(y), z \rangle|^2}{|z|^{d+\vartheta}} \mu(y) \mathrm{d}z \mathrm{d}y$ as $\vartheta \to 2^-$. Last, we prove the limit is equivalent to $2\mathcal{E}_\mu(\phi)$. For $i \neq j$, we have

$$\int_{B_1} \partial_i \phi(y) \partial_j \phi(y) z_i z_j \mathrm{d}z = - \int_{B_1} \partial_i \phi(y) \partial_j \phi(y) \tilde{z}_i \tilde{z}_j \mathrm{d}\tilde{z},$$

where $\tilde{z}_k = z_k$ for all $k \neq j$ and $\tilde{z}_j = -z_j$. Therefore, $\int_{B_1} \partial_i \phi(y) \partial_j \phi(y) z_i z_j \mathrm{d}z = 0$. As a result,

$$\int_{B_1} \frac{|\langle \nabla \phi(y), z \rangle|^2}{|z|^{d+\vartheta}} \mathrm{d}z = \int_{B_1} \frac{\sum_{i=1}^d (\partial_i \phi(y))^2 z_i^2}{|z|^{d+\vartheta}} \mathrm{d}z$$

$$= \sum_{i=1}^d (\partial_i \phi(y))^2 \frac{1}{d} \int_{B_1} \frac{|z|^2}{|z|^{d+\vartheta}} \mathrm{d}z$$

$$= |\nabla \phi(y)|^2 \frac{\pi^{\frac{d}{2}}}{(2-\vartheta)\Gamma(\frac{d}{2}+1)},$$

and the proof follows from $c_{d,\vartheta} \frac{\pi^{\frac{d}{2}}}{(2-\vartheta)\Gamma(\frac{d}{2}+1)} \to 2$ as $\vartheta \to 2^-$. $\qquad\square$

## B.2 $\chi^2$ convergence under FPI

In this section, we study the decaying property of $\chi^2$-divergence from $\rho_k^X$ to $\pi^X$, where $\rho_k^X$ is the law of $x_k$. In the following analysis, we denote $\rho_k = \rho_k^{X,Y}$ as the law of $(x_k, y_k)$, $\rho_k^Y$ the law of $y_k$. We will analyze the two steps in the stable proximal sampler separately.

**Step 1.** In the following proposition, we study the decay of $\chi^2$-divergence in step 1.

**Proposition 3.** *Assume that $\pi^X$ satisfies the $\alpha$-FPI with parameter $C_{\mathrm{FPI}(\alpha)}$, then for each $k \geq 0$,*

$$\chi^2(\rho_k^Y | \pi^Y) \leq \exp\left(-\eta \left(C_{\mathrm{FPI}(\alpha)} + \eta\right)^{-1}\right) \chi^2(\rho_k^X | \pi^X).$$

*Proof of Proposition 3.* For the simplicity of notations, we will write $p^{(\alpha)}$ and $p_t^{(\alpha)}$ as $p$ and $p_t$ respectively in this proof. Since $x_k \sim \rho_k^X$ and $y_k | x_k \sim p(\eta; x, \cdot)$, we have

$$\rho_k^Y(y) = \int_{\mathbb{R}^d} p(\eta; x, y) \rho_k^X(x) \mathrm{d}x = \int_{\mathbb{R}^d} \rho_k^X(x) p_\eta(y - x) \mathrm{d}x = \rho_k^X * p_\eta(y).$$

Therefore, we can view $\rho_k^Y$ as $\rho_k^X$ evolving along the following factional heat flow

$$\partial_t \tilde{\rho}_t = -(-\Delta)^{\frac{\alpha}{2}} \tilde{\rho}_t.$$

That is if $\tilde{\rho}_0 = \rho_k^X$, then $\tilde{\rho}_\eta = \rho_k^Y$. Similarly, since $\pi^Y = \pi^X * p_\eta$, if $\tilde{\rho}_0 = \pi^X$, then $\tilde{\rho}_\eta = \pi^Y$. For any $t \in [0, \eta]$, define $\pi_t^X = \pi^X * p_t$ and $\rho_t^X = \rho_k^X * p_t$. The derivative of $\phi$-divergence from $\rho_t^X$ to $\pi_t^X$ can be calculated as

$$\frac{d}{dt} \int_{\mathbb{R}^d} \phi(\frac{\rho_t^X}{\pi_t^X}) \pi_t^X \mathrm{d}x$$

$$= \int_{\mathbb{R}^d} \partial_t \pi_t^X \phi(\frac{\rho_t^X}{\pi_t^X}) + \phi'(\frac{\rho_t^X}{\pi_t^X}) \left( \partial_t \rho_t^X - \partial_t \pi_t^X \frac{\rho_t^X}{\pi_t^X} \right) \mathrm{d}x$$

$$= - \int_{\mathbb{R}^d} \phi(\frac{\rho_t^X}{\pi_t^X})(-\Delta)^{\frac{\alpha}{2}} \pi_t^X \mathrm{d}x + \int_{\mathbb{R}^d} \phi'(\frac{\rho_t^X}{\pi_t^X}) \left( \frac{\rho_t^X}{\pi_t^X}(-\Delta)^{\frac{\alpha}{2}} \pi_t^X - (-\Delta)^{\frac{\alpha}{2}} \rho_t^X \right) \mathrm{d}x$$

$$= \int_{\mathbb{R}^d} \left[ -\frac{\rho_t^X}{\pi_t^X}(-\Delta)^{\frac{\alpha}{2}} \phi'(\frac{\rho_t^X}{\pi_t^X}) + (-\Delta)^{\frac{\alpha}{2}} \left( \frac{\rho_t^X}{\pi_t^X} \phi'(\frac{\rho_t^X}{\pi_t^X}) \right) - (-\Delta)^{\frac{\alpha}{2}} \phi(\frac{\rho_t^X}{\pi_t^X}) \right] \pi_t^X \mathrm{d}x,$$

where in the second identity we used the distributional definition of the fractional Laplacian. Next according to the singular integral definition of fractional Laplacian, we have

$$-(-\Delta)^{\frac{\alpha}{2}} f(x) := c_{d,\alpha} \lim_{r \to 0^+} \int_{\mathbb{R}^d \setminus B_r} \frac{f(x+z) - f(x)}{|z|^{d+\alpha}} \mathrm{d}z, \tag{8}$$

where $B_r = \{x \in \mathbb{R}^d : |x| \le r\}$ and $c_{d,\alpha}$ is given in (5). With (8), we have

$$\frac{d}{dt} \int_{\mathbb{R}^d} \phi(\frac{\rho_t^X}{\pi_t^X}) \pi_t^X \mathrm{d}x$$

$$= c_{d,\alpha} \lim_{r \to 0^+} \int_{\mathbb{R}^d} \int_{\mathbb{R}^d \setminus B_r} \frac{\phi(\frac{\rho_t^X(x+z)}{\pi_t^X(x+z)}) - \phi(\frac{\rho_t^X(x)}{\pi_t^X(x)}) - \frac{\rho_t^X(x+z)}{\pi_t^X(x+z)}\phi'(\frac{\rho_t^X(x+z)}{\pi_t^X(x+z)}) + \frac{\rho_t^X(x)}{\pi_t^X(x)}\phi'(\frac{\rho_t^X(x+z)}{\pi_t^X(x+z)})}{|z|^{d+\alpha}} \mathrm{d}z \pi_t^X(x)\mathrm{d}x.$$

When $\phi(r) = (r-1)^2$, $\int_{\mathbb{R}^d} \phi(\frac{\rho_t^X}{\pi_t^X})\pi_t^X \mathrm{d}x = \chi^2(\rho_t^X|\pi_t^X)$ and we have

$$\frac{d}{dt}\chi^2(\rho_t^X|\pi_t^X) = -c_{d,\alpha} \lim_{r \to 0^+} \int_{\mathbb{R}^d} \int_{\mathbb{R}^d \setminus B_r} \frac{\left( \frac{\rho_t^X(x+z)}{\pi_t^X(x+z)} - \frac{\rho_t^X(x)}{\pi_t^X(x)} \right)^2}{|z|^{d+\alpha}} \mathrm{d}z \pi_t^X \mathrm{d}x := -\mathcal{E}_{\pi_t^X}(\frac{\rho_t^X}{\pi_t^X}).$$

According to [Cha04, Theorem 23], $p_t$ satisfies $\alpha$-FPI with parameter $t$ for all $t \in (0, \eta)$. Since $\pi^X$ also satisfies the $\alpha$-FPI with parameter $C_{\text{FPI}(\alpha)}$, Lemma 6 implies that $\pi_t^X = \pi^X * p_t$ satisfies the $\alpha$-FPI with parameter $C_{\text{FPI}(\alpha)} + \eta$ for all $t \in (0, \eta)$. Therefore we have

$$\frac{d}{dt}\chi^2(\rho_t^X|\pi_t^X) = -\mathcal{E}_{\pi_t^X}(\frac{\rho_t^X}{\pi_t^X}) \le - \left( C_{\text{FPI}(\alpha)} + \eta \right)^{-1} \chi^2(\rho_t^X|\pi_t^X).$$

Last, according to Gronwall's inequality we have

$$\chi^2(\rho_k^Y|\pi^Y) = \chi^2(\rho_\eta^X|\pi_\eta^X) \le \exp\left( -\eta \left( C_{\text{FPI}(\alpha)} + \eta \right)^{-1} \right) \chi^2(\rho_k^X|\pi^X).$$

$\square$

**Step 2.** In this step, we study the decay of $\chi^2$-divergence in step 2. building on the work by [CCSW22]. According to the R$\alpha$SO, we have $\rho_{k+1}^X(x) = \int_{\mathbb{R}^d} \pi^{X|Y}(x|y)\rho_k^Y(y)\mathrm{d}y$. Also notice that $\pi^X(x) = \int_{\mathbb{R}^d} \pi^{X|Y}(x|y)\pi^Y(y)\mathrm{d}y$. According to the data processing inequalities, $\chi^2$ divergence won't increase after step 2, i.e. $\chi^2(\rho_{k+1}^X|\pi^X) \le \chi^2(\rho_k^Y|\pi^Y)$.

Combining our results in **Step 1** and **Step 2**, we prove Theorem 3.

**Lemma 6.** *Let $\mu_1, \mu_2$ be two probability densities satisfying the $\vartheta$-FPI with parameters $C_1, C_2$ respectively. Then $\mu_1 * \mu_2$ satisfies the $\vartheta$-FPI with parameter $C_1 + C_2$.*

*Proof of Lemma 6.* Let $X, Y$ be two independent random variables such that $X \sim \mu_1$ and $Y \sim \mu_2$. Then $X + Y \sim \mu_1 * \mu_2$. According to variance decomposition, we have for any function $\phi$,

$$\text{Var}_{\mu_1 * \mu_2}(\phi) = \text{Var}(\phi(X+Y)) = \mathbb{E}\left[ \text{Var}(\phi(X+Y)|Y) \right] + \text{Var}\left( \mathbb{E}[\phi(X+Y)|Y] \right).$$

Since $X \sim \mu_1$ and $\mu_1$ satisfies the $\vartheta$-FPI with parameter $C_1$, we have

$$\text{Var}\left(\phi(X+Y)|Y\right) \leq C_1 c_{d,\alpha} \iint_{\{z \neq 0\}} \frac{\left(\phi(x+Y+z) - \phi(x+Y)\right)^2}{|z|^{(d+\vartheta)}} \mathrm{d}z \mu_1(x) \mathrm{d}x,$$

therefore we have

$$\mathbb{E}\left[\text{Var}\left(\phi(X+Y)|Y\right)\right]$$
$$\leq C_1 c_{d,\alpha} \iiint_{\{z \neq 0\}} \frac{\left(\phi(x+y+z) - \phi(x+y)\right)^2}{|z|^{(d+\vartheta)}} \mathrm{d}z \mu_1(x) \mathrm{d}x \mu_2(y) \mathrm{d}y. \tag{9}$$

Since $Y \sim \mu_2$ and $\mu_2$ satisfies the $\vartheta$-FPI with parameter $C_2$, we have

$$\text{Var}\left(\mathbb{E}\left[\phi(X+Y)|Y\right]\right)$$
$$\leq C_2 c_{d,\alpha} \iint_{\{z \neq 0\}} \frac{\left(\int \phi(x+y+z)\mu_1(x)\mathrm{d}x - \int \phi(x+y)\mu_1(x)\mathrm{d}x\right)^2}{|z|^{(d+\vartheta)}} \mathrm{d}z \mu_2(y) \mathrm{d}y$$
$$\leq C_2 c_{d,\alpha} \iint_{\{z \neq 0\}} \int \frac{\left(\phi(x+y+z) - \phi(x+y)\right)^2}{|z|^{(d+\vartheta)}} \mu_1(x) \mathrm{d}x \mathrm{d}z \mu_2(y) \mathrm{d}y \tag{10}$$

where the last inequality follows from Jensen's inequality. Combining (9) and (10), we have

$$\text{Var}_{\mu_1 * \mu_2}(\phi) \leq C_1 c_{d,\alpha} \iiint_{\{z \neq 0\}} \frac{\left(\phi(x+y+z) - \phi(x+y)\right)^2}{|z|^{(d+\vartheta)}} \mathrm{d}z \mu_1(x) \mathrm{d}x \mu_2(y) \mathrm{d}y$$
$$+ C_2 c_{d,\alpha} \iint_{\{z \neq 0\}} \int \frac{\left(\phi(x+y+z) - \phi(x+y)\right)^2}{|z|^{(d+\vartheta)}} \mu_1(x) \mathrm{d}x \mathrm{d}z \mu_2(y) \mathrm{d}y$$
$$\leq (C_1 + C_2) c_{d,\alpha} \iiint_{\{z \neq 0\}} \frac{\left(\phi(x+y+z) - \phi(x+y)\right)^2}{|z|^{(d+\vartheta)}} \mathrm{d}z \mu_1(x) \mathrm{d}x \mu_2(y) \mathrm{d}y$$
$$= (C_1 + C_2) c_{d,\alpha} \iint_{\{z \neq 0\}} \frac{\left(\phi(u+z) - \phi(u)\right)^2}{|z|^{(d+\vartheta)}} \mathrm{d}z \mu_1 * \mu_2(u) \mathrm{d}u$$
$$= (C_1 + C_2) \mathcal{E}_{\mu_1 * \mu_2}(\phi),$$

where the second inequality follows from Fatou's lemma. $\qquad\square$

### B.3 Implementation of the Stable Proximal Sampler

In this section we discuss the implementation of the R$\alpha$SO step in our stable proximal sampler. We introduce an exact implementation of the R$\alpha$SO step without optimizing the target potential and the proofs for Corollary 3 and Proposition 1.

**Rejection sampling without optimization**. Suppose a uniform lower bound of the target potential is known, i.e. there is a constant $C_{\text{Low}}$ such that $\inf_{x \in \mathbb{R}^d} V(x) \geq C_{\text{Low}} > -\infty$, R$\alpha$SO at each step can be implemented exactly via a rejection sampler with proposals $\tilde{x}_{k+1}$ following $p_\eta^{(\alpha)}(\cdot - y_k)$ and the acceptance probability $\exp(-V(\tilde{x}_{k+1}) + C_{\text{Low}})$. Then the expected number of rejections, $N$, satisfies

$$N = \left(\int_{\mathbb{R}^d} e^{-V(x)+C_{\text{Low}}} p(\eta; x, y_k) \mathrm{d}x\right)^{-1} \quad \text{and} \quad \log N = -C_{\text{Low}} - \log\left(\int_{\mathbb{R}^d} e^{-V(x)} p^{(\alpha)}(\eta; x, y_k) \mathrm{d}x\right).$$

Without loss of generality, we assume $x^* = 0$, which always hold if we translate the potential $V$ by $V(0)$. Then we have

$$\log N \leq -C_{\text{Low}} + \int_{\mathbb{R}^d} \left(V(x) - V(0)\right) p^{(\alpha)}(\eta; x, y_k) \mathrm{d}x$$
$$\leq -C_{\text{Low}} + L \int_{\mathbb{R}^d} |x + y_k|^\beta p_\eta^{(\alpha)}(x) \mathrm{d}x$$
$$\leq -C_{\text{Low}} + L\mathbb{E}_{X \sim \pi^X}[|X|^\beta] + L\eta^\beta d^{\frac{\beta}{2}} + L\mathbb{E}_{X \sim \pi^X}[|X|^{2\beta}]^{\frac{1}{2}} \chi^2(\rho_0^X | \pi^X)^{\frac{1}{2}} + \frac{\Gamma(\frac{d+1}{2})\Gamma(\frac{1-\beta}{2})L}{\Gamma(\frac{d+1-\beta}{2})\pi^{\frac{1}{2}}} \eta^\beta,$$

where the second inequality follows from Assumption 3 and the last inequality follows from the proof of Corollary 3. With the above estimation, we can pick $\eta = \Theta(C_{\text{Low}}^{\frac{1}{\beta}} d^{-\frac{1}{2}} L^{-\frac{1}{\beta}})$ and the expected number of rejections satisfies $\log N = \mathcal{O}(C_{\text{Low}} + LM)$ with $M = \mathbb{E}_{\pi^X}[|X|^\beta] + \chi^2(\rho_0^X | \pi^X) \mathbb{E}_{\pi^X}[|X|^{2\beta}]^{\frac{1}{2}}$.

*Proof of Corollary 3.* The expected number of iterations conditioned on $y_k$ in the rejection sampling is

$$N = \left( \int_{\mathbb{R}^d} e^{-V(x)+V(x^*)} p^{(\alpha)}(\eta; x, y_k) \mathrm{d}x \right)^{-1}$$

$$\text{and} \quad \log N = -V(x^*) - \log \left( \int_{\mathbb{R}^d} e^{-V(x)} p^{(\alpha)}(\eta; x, y_k) \mathrm{d}x \right)$$

$$\leq \int_{\mathbb{R}^d} \left( V(x) - V(x^*) \right) p^{(\alpha)}(\eta; x, y_k) \mathrm{d}x$$

$$= \int_{\mathbb{R}^d} \left( V(x + y_k) - V(x^*) \right) p_\eta^{(\alpha)}(x) \mathrm{d}x.$$

WLOG, assume $x^* = 0$. Since $V$ satisfies Assumption 3, we have

$$\log N \leq L \int_{\mathbb{R}^d} |x + y_k|^\beta p_\eta^{(\alpha)}(x) \mathrm{d}x = \frac{L\Gamma(\frac{d+1}{2})}{\pi^{\frac{d+1}{2}}} \eta \int_{\mathbb{R}^d} |x + y_k|^\beta (|x|^2 + \eta^2)^{-\frac{d+1}{2}} \mathrm{d}x$$

$$\leq L|y_k|^\beta + \frac{L\Gamma(\frac{d+1}{2})}{\pi^{\frac{d+1}{2}}} \eta \int_{\mathbb{R}^d} |x|^\beta (|x|^2 + \eta^2)^{-\frac{d+1}{2}} \mathrm{d}x$$

$$\leq L|y_k|^\beta + \frac{L\Gamma(\frac{d+1}{2})}{\pi^{\frac{d+1}{2}}} \eta \int_{\mathbb{R}^d} (|x|^2 + \eta^2)^{-\frac{d+1-\beta}{2}} \mathrm{d}x$$

$$= L|y_k|^\beta + \frac{\Gamma(\frac{d+1}{2})\Gamma(\frac{1-\beta}{2})L}{\Gamma(\frac{d+1-\beta}{2})\pi^{\frac{1}{2}}} \eta^\beta.$$

Therefore, when $\eta = \Theta(d^{-\frac{1}{2}} L^{-\frac{1}{\beta}})$, the expected number of rejections N is of order $\mathbb{E}[\exp(L|y_k|^\beta]$. Since $\pi^X$ satisfies a 1-FPI with parameter $C_{\text{FPI}(1)}$, according to [Cha04], $p_t$ satisfies the 1-FPI with parameter $\eta$ for any $t \in (0, \eta)$. Last it follows from Theorem 9 that for any $\eta > 0$, to achieve a $\varepsilon$-accuracy in $\chi^2$ divergence, we need to perform the stable proximal sampler $K$ steps with

$$K \geq \left( C_{\text{FPI}(1)} \eta^{-1} + 1 \right) \log \left( \frac{\chi^2(\rho_0^X | \pi^X)}{\varepsilon} \right) = \mathcal{O}\left( C_{\text{FPI}(1)} d^{\frac{1}{2}} L^{\frac{1}{\beta}} \log \left( \frac{\chi^2(\rho_0^X | \pi^X)}{\varepsilon} \right) \right).$$

$\square$

*Proof of Proposition 1.* For all $k \geq 0$, we have

$$\mathrm{TV}(\tilde{\rho}_{k+1}^X, \rho_{k+1}^X) = \mathrm{TV}\left( \int \tilde{\rho}_{k+1}^{X|Y}(\cdot|y) \tilde{\rho}_k^Y(y) \mathrm{d}y, \int \rho_{k+1}^{X|Y}(\cdot|y) \rho_k^Y(y) \mathrm{d}y \right)$$

$$\leq \mathrm{TV}\left( \int \tilde{\rho}_{k+1}^{X|Y}(\cdot|y) \tilde{\rho}_k^Y(y) \mathrm{d}y, \int \rho_{k+1}^{X|Y}(\cdot|y) \tilde{\rho}_k^Y(y) \mathrm{d}y \right)$$

$$+ \mathrm{TV}\left( \int \rho_{k+1}^{X|Y}(\cdot|y) \tilde{\rho}_k^Y(y) \mathrm{d}y, \int \rho_{k+1}^{X|Y}(\cdot|y) \rho_k^Y(y) \mathrm{d}y \right)$$

$$\leq \mathbb{E}_{\tilde{\rho}_k^Y}[\mathrm{TV}(\tilde{\rho}_{k+1}^{X|Y}(\cdot, y)], \rho_{k+1}^{X|Y}(\cdot|y)) + \mathrm{TV}(\tilde{\rho}_k^Y, \rho_k^Y)$$

$$\leq \varepsilon_{\mathrm{TV}} + \mathrm{TV}(\tilde{\rho}_k^X, \rho_k^X),$$

where the last two inequalities follow from the data processing inequality. Therefore, $\mathrm{TV}(\tilde{\rho}_k^X, \rho_k^X) \leq k\varepsilon_{\mathrm{TV}} + \mathrm{TV}(\tilde{\rho}_0^X, \rho_0^X)$ for all $k \geq 1$.

Next, the iteration complexity of Algorithm 2 with an inexact R$\alpha$SO can be obtained from Proposition 1. Since $\tilde{\rho}_0^X = \rho_0^X$, according to Pinsker's inequality, we have

$$\mathrm{TV}(\tilde{\rho}_k^X, \pi^X) \leq \mathrm{TV}(\tilde{\rho}_k^X, \rho_k^X) + \mathrm{TV}(\rho_k^X, \pi^X) \leq \mathrm{TV}(\tilde{\rho}_k^X, \rho_k^X) + \sqrt{\chi^2(\rho_k^X|\pi^X)/2}$$

$$\leq k\varepsilon_{\mathrm{TV}} + \sqrt{\exp(-k\eta(C_{\mathrm{FPI}(\alpha)} + \eta)^{-1})\chi^2(\tilde{\rho}_0^X|\pi^X)/2}.$$

For any $\varepsilon > 0$ and any $K$ satisfies

$$K \geq (C_{\mathrm{FPI}(\alpha)}\eta^{-1} + 1)\ln\left(2\chi^2(\tilde{\rho}_0^X|\pi^X)/\varepsilon^2\right),$$

if the R$\alpha$SO can be implemented inexactly with $\varepsilon_{\mathrm{TV}} \leq \frac{\varepsilon}{2K}$, the density of the $K^{\mathrm{th}}$ iterate of Algorithm 2 is $\varepsilon$-close to the target in the total variation distance, i.e. $\mathrm{TV}(\tilde{\rho}_X^K, \pi^X) \leq \varepsilon$. $\qquad\square$

### B.4 Convergence under Weak Fractional Poincaré Inequality

Our main result for Algorithm 2 in Theorem 3 is proved under the assumption the target satisfying $\alpha$-FPI. Furthermore, for the rejection-sampling based implementation of the R$\alpha$SO in Algorithm 3, the parameter $\alpha$ is set to be 1. In order to use Theorem 3 for the case of generalized Cauchy targets, one has to check if the $\alpha$-FPI is satisfied or not, which depends on the degrees of freedom parameter $\nu$ of the generalized Cauchy desity. Specifically, when $\nu \geq 1$, 1-FPI is satisfied and we hence have Corollary 5, part (i) based on Theorem 3. When $\nu \in (0, 1)$, 1-FPI is not satisfied and hence Theorem 3 no longer applies.

To tackle this issue, we now introduce a generalization of Theorem 3 to the case when the target satisfies a weak version of Fractioanl Poincaré inequality (wFPI) and provide convergence guarantees for the stable proximal sampler in $\chi^2$-divergence.

**Definition 3** (weak Fractional Poincaré Inequality). *For $\vartheta \in (0, 2)$, a probability density $\mu$ satisfies a $\vartheta$-weak fractional Poincaré inequality if there exists a decreasing function $\beta_{\mathrm{WFPI}(\vartheta)} : \mathbb{R}_+ \to \mathbb{R}_+$ such that for any $\phi : \mathbb{R}^d \to \mathbb{R}$ in the domain of $\mathcal{E}_\mu^{(\vartheta)}$ with $\mu(\phi) = 0$, we have*

$$\mu(\phi^2) \leq \beta_{\mathrm{WFPI}(\vartheta)}(r)\mathcal{E}_\mu^{(\vartheta)}(\phi) + r\|\phi\|_\infty^2, \qquad \forall r > 0, \tag{wFPI}$$

*where $\mathcal{E}_\mu^{(\vartheta)}$ is a non-local Dirichlet form associated with $\mu$ defined as*

$$\mathcal{E}_\mu^{(\vartheta)}(\phi) := c_{d,\vartheta} \iint_{\{x \neq y\}} \frac{(\phi(x) - \phi(y))^2}{|x - y|^{(d+\vartheta)}} \mathrm{d}x\mu(y)\mathrm{d}y \quad \text{with} \quad c_{d,\vartheta} = \frac{2^\vartheta \Gamma((d+\vartheta)/2)}{\pi^{d/2}|\Gamma(-\vartheta/2)|}.$$

The wFPI is satisfied by any probability density that is locally bounded, and is hence extremely general. Setting the parameter $r = 0$, wFPI reduces to FPI with $C_{\mathrm{FPI}(\vartheta)} = \beta_{\mathrm{WFPI}(\vartheta)}(0)$.

**Theorem 5.** *Assume that $\pi^X$ satisfies the $\alpha$-wFPI with parameter $\beta_{\mathrm{WFPI}(\alpha)}(r)$ for some $\alpha \in (0, 2)$. Then for any step size $\eta > 0$ and initial condition $\rho_0^X$ such that $R_\infty(\rho_0^X|\pi^X) < \infty$, the $k^{th}$ iterate of the stable proximal sampler with parameter $\alpha$ (Algorithm 2) satisfies*

$$\chi^2(\rho_k^X|\pi^X) \leq \exp\left(-(\beta_{\mathrm{WFPI}(\alpha)}(r) + \eta)^{-1}k\eta\right)\chi^2(\rho_0^X|\pi^X)$$
$$+ 4r\left(1 - \exp\left(-(\beta_{\mathrm{WFPI}(\alpha)}(r) + \eta)^{-1}(k+1)\eta\right)\right)\exp\left(2R_\infty(\rho_0^X|\pi^X)\right).$$

The proof of Theorem 5 follows the same two-step analysis as it is introduced in the beginning of Section B.2. The convergence property corresponding to **Step 1** is stated in the following Proposition.

**Proposition 4.** *Assume that $\pi^X$ satisfies the $\alpha$-wFPI with parameter $\beta_{\mathrm{WFPI}(\alpha)}$ for some $\alpha \in (0, 2)$, then for each $k \geq 0, r > 0$,*

$$\chi^2(\rho_k^Y|\pi^Y) \leq \exp\left(-(\beta_{\mathrm{WFPI}(\alpha)}(r) + \eta)^{-1}\eta\right)\chi^2(\rho_k^X|\pi^X)$$
$$+ 4r\left(1 - \exp\left(-(\beta_{\mathrm{WFPI}(\alpha)}(r) + \eta)^{-1}\eta\right)\right)\exp\left(2R_\infty(\rho_k^X|\pi^X)\right). \tag{11}$$

*Proof of Proposition 4.* In the stable proximal sampler with parameter $\alpha$, we have $\rho_k^Y = \rho_k^X * p_\eta^{(\alpha)}$ and $\pi^Y = \pi^X * p_\eta^{(\alpha)}$. Therefore we can view $\rho_k^Y$ and $\pi^Y$ as $\rho_k^X$ and $\pi^X$ evolving along the fractional

heat flow by time $\eta$ respectively. For any $t \in [0, \eta]$, define $\pi_t^X = \pi^X * p_t^{(\alpha)}$ and $\rho_t^X = \rho_k^X * p_t^{(\alpha)}$. We have

$$\frac{d}{dt}\chi^2(\rho_t^X|\pi_t^X) = -\mathcal{E}_{\pi_t^X}\left(\frac{\rho_t^X}{\pi_t^X}\right) = -\mathcal{E}_{\pi_t^X}\left(\frac{\rho_t^X}{\pi_t^X} - 1\right).$$

According to [Cha04, Theorem 23], $p_t^{(\alpha)}$ satisfies $\alpha$-FPI with parameter $\eta$ for all $t \in (0, \eta]$. According to Lemma 7, $\pi_t^X$ satisfies the $\alpha$-wFPI with $\beta_{\text{WFPI}(\alpha)}(r) + \eta$. Therefore we get

$$\frac{d}{dt}\chi^2(\rho_t^X|\pi_t^X) \leq \left(\beta_{\text{WFPI}(\alpha)}(r) + \eta\right)^{-1}\chi^2(\rho_t^X|\pi_t^X) + r\left(\beta_{\text{WFPI}(\alpha)}(r) + \eta\right)^{-1}\left\|\rho_t^X/\pi_t^X - 1\right\|_\infty^2$$

$$\leq \left(\beta_{\text{WFPI}(\alpha)}(r) + \eta\right)^{-1}\chi^2(\rho_t^X|\pi_t^X) + 4r\left(\beta_{\text{WFPI}(\alpha)}(r) + \eta\right)^{-1}\exp\left(2R_\infty(\rho_k^X|\pi^X)\right),$$

where the last inequality follows from the definition of Renyi-divergence and the data processing inequality. Last, (11) follows from Gronwall's inequality. $\qquad\square$

*Proof of Theorem 5.* According to Proposition 4, the $\chi^2$ decaying property in step 1 of the algorithm is as follows,

$$\chi^2(\rho_k^Y|\pi^Y) \leq \exp\left(-\left(\beta_{\text{WFPI}(\alpha)}(r) + \eta\right)^{-1}\eta\right)\chi^2(\rho_k^X|\pi^X)$$
$$+ 4r\left(1 - \exp\left(-\left(\beta_{\text{WFPI}(\alpha)}(r) + \eta\right)^{-1}\eta\right)\right)\exp\left(2R_\infty(\rho_k^X|\pi^X)\right).$$

In step 2, we have $\rho_{k+1}^X = \rho_k^Y * \pi^{X|Y}$ and $\pi^X = \pi^Y * \pi^{X|Y}$. Therefore according to the data processing inequality, we get

$$\chi^2(\rho_{k+1}^X|\pi^X) \leq \chi^2(\rho_k^Y|\pi^Y)$$
$$\leq \exp\left(-\left(\beta_{\text{WFPI}(\vartheta)}(r) + \eta\right)^{-1}\eta\right)\chi^2(\rho_k^X|\pi^X)$$
$$+ 4r\left(1 - \exp\left(-\left(\beta_{\text{WFPI}(\alpha)}(r) + \eta\right)^{-1}\eta\right)\right)\exp\left(2R_\infty(\rho_k^X|\pi^X)\right)$$
$$\leq \exp\left(-k(\beta_{\text{WFPI}(\alpha)}(r) + \eta)^{-1}\eta\right)\chi^2(\rho_0^X|\pi^X)$$
$$+ +4r\left(1 - \exp\left(-\left(\beta_{\text{WFPI}(\alpha)}(r) + \eta\right)^{-1}(k+1)\eta\right)\right)\exp\left(2R_\infty(\rho_0^X|\pi^X)\right),$$

where the last inequality follows from the data processing inequality. Last, apply the above iterative relation $k$ times and we prove (11). $\qquad\square$

**Lemma 7.** *Let $\mu_1$ be a probability density on $\mathbb{R}^d$ satisfying the $\vartheta$-wFPI with parameter $\beta_{\text{WFPI}(\vartheta)}(r)$. Let $\mu_2$ be a probability density on $\mathbb{R}^d$ satisfying the $\vartheta$-FPI with parameter $C_{\text{FPI}(\vartheta)}$. Then $\mu_1 * \mu_2$ satisfies $\vartheta$-wFPI with parameter $\beta_{\text{WFPI}(\vartheta)}(r) + C_{\text{FPI}(\vartheta)}$.*

*Proof of Lemma 7.* Let $X, Y$ be two independent random variables such that $X \sim \mu_2$ and $Y \sim \mu_1$. According to variance decomposition, we have for any function $\phi$ such that $\mu_1 * \mu_2(\phi) = 0$,

$$\text{Var}_{\mu_1 * \mu_2}(\phi) = \text{Var}\left(\phi(X + Y)\right) = \mathbb{E}\left[\text{Var}\left(\phi(X + Y)|Y\right)\right] + \text{Var}\left(\mathbb{E}\left[\phi(X + Y)|Y\right]\right).$$

Since $X \sim \mu_2$ and $\mu_2$ satisfies the $\vartheta$-FPI with parameter $C_{\text{FPI}(\vartheta)}$, we have

$$\mathbb{E}[\text{Var}\left(\phi(X + Y)|Y\right)] \tag{12}$$

$$\leq C_{\text{FPI}(\vartheta)}c_{d,\alpha}\iiint_{\{z \neq 0\}}\frac{(\phi(x + y + z) - \phi(x + y))^2}{|z|^{(d+\vartheta)}}dz\mu_2(x)dx\mu_1(y)dy. \tag{13}$$

Since $Y \sim \mu_1$ and $\mu_1$ satisfies the $\vartheta$-wFPI with parameter $\beta_{\text{WFPI}(\vartheta)}$, following the proof of Lemma 6, we have

$$\text{Var}\left(\mathbb{E}\left[\phi(X + Y)|Y\right]\right)$$

$$\leq \beta_{\text{WFPI}(\vartheta)}c_{d,\alpha}\iint_{\{z \neq 0\}}\int\frac{(\phi(x + y + z) - \phi(x + y))^2}{|z|^{(d+\vartheta)}}\mu_2(x)dxdz\mu_1(y)dy$$

$$+ r\left\|\int\phi(x + \cdot)\mu_2(x)dx - \iint\phi(x + y)\mu_2(x)dx\mu_1(y)dy\right\|_\infty^2 \tag{14}$$

$$\leq \beta_{\text{WFPI}(\vartheta)}c_{d,\alpha}\iint_{\{z \neq 0\}}\int\frac{(\phi(x + y + z) - \phi(x + y))^2}{|z|^{(d+\vartheta)}}\mu_2(x)dxdz\mu_1(y)dy + r\|\phi\|_\infty^2,$$

where the last inequality follows from the fact that $\mu_1 * \mu_2(\phi) = 0$ and the convexity $\|\cdot\|_\infty$. Combining (12) and (14), we have

$$\mathrm{Var}_{\mu_1 * \mu_2}(\phi)$$

$$\leq C_{\mathrm{FPI}(\vartheta)} c_{d,\alpha} \iiint_{\{z \neq 0\}} \frac{(\phi(x+y+z) - \phi(x+y))^2}{|z|^{(d+\vartheta)}} \mathrm{d}z \mu_2(x) \mathrm{d}x \mu_1(y) \mathrm{d}y$$

$$+ \beta_{\mathrm{WFPI}(\vartheta)}(r) c_{d,\alpha} \iint_{\{z \neq 0\}} \int \frac{(\phi(x+y+z) - \phi(x+y))^2}{|z|^{(d+\vartheta)}} \mu_2(x) \mathrm{d}x \mathrm{d}z \mu_1(y) \mathrm{d}y + r \|\phi\|_\infty^2$$

$$= \left(\beta_{\mathrm{WFPI}(\vartheta)}(r) + C_{\mathrm{FPI}(\vartheta)}\right) c_{d,\alpha} \iint_{\{z \neq 0\}} \frac{(\phi(u+z) - \phi(u))^2}{|z|^{(d+\vartheta)}} \mathrm{d}z \mu_1 * \mu_2(u) \mathrm{d}u + r \|\phi\|_\infty^2$$

$$= \left(\beta_{\mathrm{WFPI}(\vartheta)}(r) + C_{\mathrm{FPI}(\vartheta)}\right) \mathcal{E}_{\mu_1 * \mu_2}(\phi) + r \|\phi\|_\infty^2 .$$

Lemma 7 is hence proved. $\qquad\square$

## B.5 Proofs for the Generalized Cauchy Examples

In this section, we provide proofs for the two corollaries in Section 3.2.

*Proof of Corollary 4.* According to [WW15, Corollary 1.2], $\pi_\nu$ satisfies $\alpha$-FPI with parameter $C_{\mathrm{FPI}(\vartheta)}$ for any $\alpha \leq \min(2, \nu)$. Therefore it follows from Theorem 3 that

$$\chi^2(\rho_k^X | \pi_\nu) \leq \exp\left(-k\eta \left(C_{\mathrm{FPI}(\alpha)} + \eta\right)^{-1}\right) \chi^2(\rho_0^X | \pi_\nu). \tag{15}$$

According to [MHFH$^+$23, Corollary 22], when $\rho_0^X = \mathcal{N}(0, I_d)$ and $d \geq 2$, $R_\infty(\rho_0^X | \pi_\nu) \leq \ln(2^{\nu/2}\Gamma(\nu/2)) + \ln(\frac{d+\nu}{2e})$ which implies $\chi^2(\rho_0^X | \pi_\nu) = \Theta(d)$. Therefore Corollary 4 follows from (15) and $\eta \in (0, 1)$. $\qquad\square$

*Proof of Corollary 5.* We prove the two part in the Corollary separately:

**(i) When $\nu \geq 1$,** according to [WW15, Corollary 1.2] $\pi_\nu$ satisfies the 1-FPI with parameter $C_{\mathrm{FPI}(1)}$. Corollary 3 applies with $L = 4(d+\nu)$ and $\beta = 1/4$ and the iteration complexity of Algorithm 2 is of order $\mathcal{O}\left(C_{\mathrm{FPI}(1)} d^{\frac{1}{2}} (d+\nu)^4 \ln(\chi^2(\rho_0^X | \pi_\nu)/\varepsilon)\right)$.

**(ii) When $\nu \in (0, 1)$,** according to [WW15, Corollary 1.2], there exists a positive constant $c$ such that $\pi_\nu$ satisfies the 1-wFPI with parameter

$$\beta_{\mathrm{WFPI}(1)}(r) = c(1 + r^{-(1-\nu)/\nu}). \tag{16}$$

Theorem 5 implies that

$$\chi^2(\rho_k^X | \pi_\nu) \leq \exp\left(-\frac{k\eta}{\eta + c(1 + r^{-(1-\nu)/\nu})}\right) \chi^2(\rho_0^X | \pi_\nu)$$

$$+ r\left(1 - \exp\left(-\frac{(k+1)\eta}{\eta + c(1 + r^{-(1-\nu)/\nu})}\right)\right) \exp\left(2R_\infty(\rho_0^X | \pi^X)\right)$$

$$\leq \exp\left(-\frac{k\eta}{\eta + c(1 + r^{-(1-\nu)/\nu})}\right) \chi^2(\rho_0^X | \pi_\nu)$$

$$+ \frac{(k+1)\eta r}{\eta + c(1 + r^{-(1-\nu)/\nu})} \exp\left(2R_\infty(\rho_0^X | \pi^X)\right).$$

For any $\varepsilon > 0$ and $k \geq 1$, pick $r = \frac{\exp\left(-2\nu R_\infty(\rho_0^X | \pi_\nu)\right) c^\nu \varepsilon^\nu}{(k+1)^\nu \eta^\nu}$, we have $\chi^2(\rho_k^X | \pi_\nu) \leq \varepsilon$ if

$$k \geq \left[1 + c^{\frac{1}{\nu}} \eta^{-\frac{1}{\nu}} + 2^{1/\nu} c\eta^{-1} \varepsilon^{-(1-\nu)/\nu} \exp\left(\frac{2(1-\nu)R_\infty(\rho_0^X | \pi_\nu)}{\nu}\right)\right] \ln^{1/\nu}\left(\frac{2\chi^2(\rho_0^X | \pi_\nu)}{\varepsilon}\right).$$

Corollary 3 applies with $L = (d+\nu)/\nu$ and $\beta = \nu/4$. Therefore, by choosing $\eta = \Theta(d^{-\frac{1}{2}}(d+\nu)^{-\frac{4}{\nu}})$, the iteration complexity in Algorithm 2 is of order

$$\mathcal{O}\left(\max\left\{c^{\frac{1}{\nu}} d^{\frac{1}{2\nu} + \frac{4}{\nu^2}}, cd^{\frac{1}{2} + \frac{4}{\nu}} \varepsilon^{-\frac{1-\nu}{\nu}} \exp\left(\frac{2(1-\nu)R_\infty(\rho_0^X | \pi_\nu)}{\nu}\right)\right\} \ln^{\frac{1}{\nu}}\left(\frac{2\chi^2(\rho_0^X | \pi_\nu)}{\varepsilon}\right)\right).$$

$\qquad\square$

# C  Proofs for the Lower Bounds on the Stable Proximal Sampler

In this section we introduce the proofs for the lower bounds for the stable proximal sampler with parameter $\alpha$ when the target is the generalized Cauchy density with degrees of freedom strictly smaller than $\alpha$. The lower bound is proved following the idea introduced in Section 2.

**Lemma 8.** *Suppose $(x_k, y_k)_k$ are the iterates of the stable proximal sampler with parameter $\alpha$, step size $\eta$ and target density $\pi^X \propto \exp(-V)$ for some $V : \mathbb{R}^d \to \mathbb{R}$. Let $G(x) = \exp(\kappa V(x))$ with $\kappa \in (0, 1)$. Then, for every $k \geq 0$,*

$$\mathbb{E}[G(x_{k+1})] \leq \mathbb{E}[G(x_k + 2^{\frac{1}{\alpha}} \eta^{\frac{1}{\alpha}} z_k)],$$

*where $z_k$, with density $p_1^{(\alpha)}$, is sampled independently from $x_k$.*

*Proof of Lemma 8.* Recall that $\pi^{X|Y}(x|y) \propto \pi^X(x)p^{(\alpha)}(\eta; x, y)$. We have

$$\mathbb{E}[G(x_{k+1})] = \mathbb{E}\big[\mathbb{E}[G(x_{k+1})|y_k]\big] = \mathbb{E}\Big[Z_{y_k}^{-1} \int G(x)\pi^X(x)p_\eta^{(\alpha)}(x - y_k)\mathrm{d}x\Big]$$

$$= \mathbb{E}\big[Z_{y_k}^{-1} \mathbb{E}[G(y_k + \eta^{\frac{1}{\alpha}} z_k)\pi^X(y_k + \eta^{\frac{1}{\alpha}} z_k)|y_k]\big],$$

where $Z_{y_k} = \int \pi^X(x)p_\eta^{(\alpha)}(x - y_k)\mathrm{d}x = \mathbb{E}[\pi^X(y_k + \eta^{\frac{1}{\alpha}} z_k)|y_k]$ and $z_k$ is the $\alpha$-stable random vector with density $p_1^{(\alpha)}$, which is independent to $y_k, x_k$. Let $T : \mathbb{R}_+ \to \mathbb{R}$ be $T(r) = r^{-\kappa}$. Since $\kappa \in (0, 1)$, $T$ is convex and $r \mapsto rT(r)$ is concave. According to the fact that $G(x) = T(\pi^X)(x)$ and Jensen's inequality, we have

$$\mathbb{E}[G(x_{k+1})] = \mathbb{E}\left[\frac{\mathbb{E}\big[(\pi^X T(\pi^X))(y_k + \eta^{\frac{1}{\alpha}} z_k)|y_k\big]}{\mathbb{E}\big[\pi^X(y_k + \eta^{\frac{1}{\alpha}} z_k)|y_k\big]}\right]$$

$$\leq \mathbb{E}\big[T\big(\mathbb{E}[\pi^X(y_k + \eta^{\frac{1}{\alpha}} z_k)|y_k]\big)\big].$$

Since $T$ is convex, apply Jensen's inequality again and we get

$$\mathbb{E}[G(x_{k+1})] \leq \mathbb{E}[G(y_k + \eta^{\frac{1}{\alpha}} z_k)] = \mathbb{E}\big[\mathbb{E}[G(x_k + \eta^{\frac{1}{\alpha}} z'_k + \eta^{\frac{1}{\alpha}} z_k)|x_k]\big]$$

$$= \mathbb{E}[G(x_k + 2^{\frac{1}{\alpha}} \eta^{\frac{1}{\alpha}} \bar{z}_k)|x_k],$$

where $z'_k$ is the $\alpha$-stable random vector with density $p_1^{(\alpha)}$, which is independent to $x_k, z_k$ and the last identity follows from the self-similarity of $\alpha$-stable process with $\bar{z}_k \sim p_1^{(\alpha)}$ which is independent to $x_k$. $\qquad\square$

**Lemma 9.** *Suppose $(x_k, y_k)_k$ are the iterates of the stable proximal sampler with parameter $\alpha$, step size $\eta$ and target density $\pi^X \propto \exp(-V)$ satisfies*

$$|\nabla V(x)| \leq \frac{(d + \nu_2)|x|}{1 + |x|^2} \quad \text{and} \quad \Delta V(x) \leq \frac{(d + \nu_2)^2}{1 + |x|^2},$$

*for some $\nu_2 \in (0, \alpha)$ and for all $x \in \mathbb{R}^d$. Let $G(x) = \exp(\kappa V(x))$ with*

$$\kappa \in (\nu_2(d + \nu_2)^{-1}, \alpha(d + \nu_2)^{-1}).$$

*Then, for every $k \geq 0$ and for all $r > 0$,*

$$\mathbb{E}[G(x_{k+1})] \leq (1 + r)^{\frac{\kappa(d+\nu_2)}{2}} \mathbb{E}[G(x_k)] + 2^{\frac{\kappa(d+\nu_2)}{\alpha}} \eta^{\frac{\kappa(d+\nu_2)}{\alpha}} (1 + r^{-1})^{\frac{\kappa(d+\nu_2)}{2}} m_{\kappa(d+\nu_2)}^{(\alpha)}, \quad (17)$$

*where $m_{\kappa(d+\nu_2)}^{(\alpha)} = \mathbb{E}[|z_k|^{\kappa(d+\nu_2)}]$ with $z_k$ being an $\alpha$-stable random vector with density $p_1^{(\alpha)}$. Moreover, for every $N \geq 0$,*

$$\mathbb{E}[G(x_N)] \lesssim \mathbb{E}[G(x_0)] + m_{\kappa(d+\nu_2)}^{(\alpha)} N^{\frac{\kappa(d+\nu_2)}{2}+1} \eta^{\frac{\kappa(d+\nu_2)}{\alpha}}, \quad (18)$$

*where $\lesssim$ is hiding a uniform positive constant factor.*

*Proof of Lemma 9.* Without loss of generality assume $V(0) = 0$. Then, we have that,

$$V(x) = \int_0^1 \langle x, \nabla V(tx)\rangle \mathrm{d}t \le (d+\nu_2)\int_0^1 \frac{t|x|}{1+|tx|^2}\mathrm{d}t = \frac{d+\nu_2}{2}\ln(1+|x|^2).$$

Therefore $G(x) = \exp(\kappa V(x)) \le (1+|x|^2)^{\kappa(d+\nu_2)/2}$, Since $\kappa \in (\nu_2(d+\nu_2)^{-1}, \alpha(d+\nu_2)^{-1})$, $G(x) = \mathcal{O}(|x|^{\kappa(d+\nu_2)})$ when $|x| \gg 1$ and $\mathbb{E}[G(x_k + 2^{\frac{1}{\alpha}}\eta^{\frac{1}{\alpha}}z_k)]$ in Lemma 8 is finite. We have

$$\mathbb{E}[G(x_k + 2^{\frac{1}{\alpha}}\eta^{\frac{1}{\alpha}}z_k)]$$

$$\le \mathbb{E}[(1+|x_k + 2^{\frac{1}{\alpha}}\eta^{\frac{1}{\alpha}}z_k|^2)^{\frac{\kappa(d+\nu_2)}{2}}]$$

$$\le \mathbb{E}[(1+(1+r)|x_k|^2 + 4^{\frac{1}{\alpha}}\eta^{\frac{2}{\alpha}}(1+r^{-1})|z_k|^2)^{\frac{\kappa(d+\nu_2)}{2}}]$$

$$\le (1+r)^{\frac{\kappa(d+\nu_2)}{2}}\mathbb{E}[G(x_k)] + 2^{\frac{\kappa(d+\nu_2)}{\alpha}}\eta^{\frac{\kappa(d+\nu_2)}{\alpha}}(1+r^{-1})^{\frac{\kappa(d+\nu_2)}{2}}\mathbb{E}[|z_k|^{\kappa(d+\nu_2)}]$$

$$\le (1+r)^{\frac{\kappa(d+\nu_2)}{2}}\mathbb{E}[G(x_k)] + 2^{\frac{\kappa(d+\nu_2)}{\alpha}}\eta^{\frac{\kappa(d+\nu_2)}{\alpha}}(1+r^{-1})^{\frac{\kappa(d+\nu_2)}{2}}m^{(\alpha)}_{\kappa(d+\nu_2)},$$

where the first inequality follows from the Young's inequality and $m^{(\alpha)}_{\kappa(d+\nu_2)} = \mathbb{E}[|z_k|^{\kappa(d+\nu_2)}]$ with $z_k$ being an $\alpha$-stable random vector with density $p_1^{(\alpha)}$. (17) follows from Lemma 8. Furthermore, by induction we have

$$\mathbb{E}[G(x_N)] \le (1+r)^{\kappa(d+\nu_2)N/2}\mathbb{E}[G(x_0)]$$

$$+ \frac{(1+r)^{\kappa(d+\nu_2)N/2}-1}{(1+r)^{\kappa(d+\nu_2)/2}-1}2^{\frac{\kappa(d+\nu_2)}{\alpha}}\eta^{\frac{\kappa(d+\nu_2)}{\alpha}}(1+r^{-1})^{\frac{\kappa(d+\nu_2)}{2}}m^{(\alpha)}_{\kappa(d+\nu_2)}.$$

Pick $r = \frac{2}{\kappa(d+\nu_2)N}$ and (18) is proved. $\qquad\qquad\qquad\qquad\qquad\qquad\qquad\square$

*Proof of Theorem 4.* To apply Lemma 1, we choose $G(x) = \exp(\kappa V(x))$ with $\kappa \in (\nu_2(d+\nu_2)^{-1}, \alpha(d+\nu_2)^{-1}) \subset (0,1)$. Without loss of generality assume $V(0) = 0$. Via Assumption 1, we have the estimates for $V$,

$$V(x) = \int_0^1 \langle x, \nabla V(tx)\rangle \mathrm{d}t \ge (d+\nu_1)\int_0^1 \frac{t|x|}{1+|tx|^2}\mathrm{d}t = \frac{d+\nu_1}{2}\ln(1+|x|^2).$$

By Lemma 2 we have

$$\pi^X(G(x) \ge y) \ge \pi^X\left(|x| \ge y^{\frac{1}{\kappa(d+\nu_1)}}\right) \ge C_{\nu_1}d^{\frac{\nu_1}{2}}\left(1+y^{\frac{-2}{\kappa(d+\nu_1)}}\right)^{-\frac{d+\nu_2}{2}}y^{\frac{-\nu_2}{\kappa(d+\nu_1)}}.$$

We then invoke Lemma 1 and Lemma 9 to obtain

$$\mathrm{TV}(\rho_N^X, \pi^X)$$

$$\gtrsim \sup_{y\ge 1}C_{\nu_1}d^{\frac{\nu_1}{2}}\left(1+y^{\frac{-2}{\kappa(d+\nu_1)}}\right)^{-\frac{d+\nu_2}{2}}y^{\frac{-\nu_2}{\kappa(d+\nu_1)}} - \frac{\mathbb{E}[G(x_0)]+m^{(\alpha)}_{\kappa(d+\nu_2)}N^{\frac{\kappa(d+\nu_2)}{2}+1}\eta^{\frac{\kappa(d+\nu_2)}{\alpha}}}{y}.$$

The fact that $\kappa \in (\nu_2(d+\nu_1)^{-1}, \alpha(d+\nu_2)^{-1})$ ensures that the supremum on the right side is always positive. In particular, picking $y$ such that

$$y^{1-\frac{\nu_2}{\kappa(d+\nu_1)}} = \Theta\left(C_{\nu_1}^{-1}d^{-\frac{\nu_2}{2}}\left(\mathbb{E}[G(x_0)]+m^{(\alpha)}_{\kappa(d+\nu_2)}N^{\frac{\kappa(d+\nu_2)}{2}+1}\eta^{\frac{\kappa(d+\nu_2)}{\alpha}}\right)\right),$$

we obtain that

$$\mathrm{TV}(\rho_N^X, \pi^X)$$

$$\gtrsim C_{\nu_1}^{\frac{\kappa(d+\nu_1)}{\kappa(d+\nu_1)-\nu_2}}d^{\frac{\kappa(d+\nu_1)\nu_2}{2\kappa(d+\nu_1)-2\nu_2}}\left(\mathbb{E}[G(x_0)]+m^{(\alpha)}_{\kappa(d+\nu_2)}N^{\frac{\kappa(d+\nu_2)}{2}+1}\eta^{\frac{\kappa(d+\nu_2)}{\alpha}}\right)^{-\frac{\nu_2}{\kappa(d+\nu_1)-\nu_2}},$$

where $\gtrsim$ is hiding a uniform positive constant factor. Therefore, for any $\alpha \in (\frac{\nu_2(d+\nu_2)}{d+\nu_1}, 2]$ and $\delta \in (0, \alpha - \frac{\nu_2(d+\nu_2)}{d+\nu_1})$, we can choose $\kappa = \frac{\alpha-\delta}{d+\nu_2} \in (\frac{\nu_2}{d+\nu_1}, \frac{\alpha}{d+\nu_2})$ and get that

$$\mathrm{TV}(\rho_N^X, \pi^X)$$

$$\ge C_{\nu_1,\nu_2,\delta}d^{\frac{\nu_2(\alpha-\delta)(d+\nu_1)}{2(\alpha-\delta)(d+\nu_1)-2\nu_2(d+\nu_2)}}\left(\mathbb{E}[G(x_0)]+m^{(\alpha)}_{\alpha-\delta}N^{\frac{\alpha-\delta}{2}+1}\eta^{\frac{\alpha-\delta}{\alpha}}\right)^{-\frac{\nu_2(d+\nu_2)}{(\alpha-\delta)(d+\nu_1)-\nu_2(d+\nu_2)}}.$$

Theorem 4 then follows by taking $\tau = \alpha - \delta$. $\qquad\qquad\qquad\qquad\qquad\qquad\square$

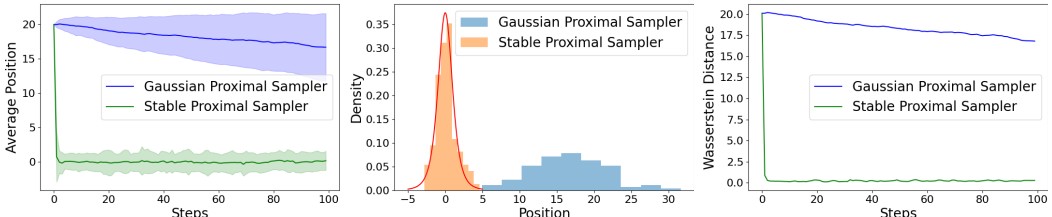

Figure 1: Comparison between Gaussian and Stable Proximal Sampler: target is chosen to be one-dimensional student-t with center $0$ and $4$ degrees of freedom; initialization is chosen $x_0 = 20$.

## C.1 Further Discussions on Lower bounds of the stable proximal sampler

To derive a lower bound for the stable proximal sampler with parameter $\alpha$, it is worth mentioning that there is an extra difficulty applying our method when $\nu \geq \alpha$. Recall that when $\nu \in (0, \alpha)$, $\pi_\nu$ has heavier tail than $\rho_k^X$ does. Therefore, when we apply

$$\text{TV}(\rho_k^X, \pi_\nu) \geq |\pi_\nu(G \geq y) - \rho_k^X(G \geq y)|, \tag{19}$$

to study the lower bound, it suffices to derive a lower bound on $\pi_\nu(G \geq y)$, and an upper bound on $\rho_k^X(G \geq y)$ which is smaller than the lower bound on $\pi_\nu(G \geq y)$. Deriving these bounds is not too hard: the lower bound can be obtained by looking at an explicit integral against $\pi_\nu$ directly and the upper bound is derived based on the fractional absolute moment accumulation of the isotropic $\alpha$-stable random variables along the stable proximal sampler.

However, when $\nu \geq \alpha$, we expect that $\rho_k^X$ has heavier tail than $\pi_\nu$. Therefore, to apply (19), we need to find an upper bound on $\pi_\nu(G \geq y)$, and a lower bound on $\rho_k^X(G \geq y)$ which is smaller than the upper bound on $\pi_\nu(G \geq y)$. Notice that $\rho_k^X(G \geq y)$ is a quantity varying along the trajectory of the stable proximal sampler. Deriving a lower bound along the trajectory is essentially more challenging than deriving an upper bound.

In order to derive a satisfying lower bound in this case, it hence remains to characterize the stable proximal sampler as an approximation of an appropriate gradient flow, just as that the Brownian-driven proximal sampler can be interpret as the entropy-regularized JKO scheme in [CCSW22]; see also Section 5. To understand this kind of gradient flow approximations itself is an interesting future work as it may help us to understand and characterize the class of MCMC samplers that utilize heavy-tail samples to approximate lighter-tail target densities, which is non-standard compared to commonly used MCMC samplers such as ULA, MALA, etc.

## D  Numerical Illustrations

In this section, we present numerical results that illustrate the improved performance of the proximal sampler with stable oracles ($\alpha = 1$) compared to that with Gaussian oracles. We first sample from the one-dimensional student-t distribution with center zero and $4$ degrees of freedom by running the proximal samplers with different oracles in parallel for 100 times. Each individual chain is run for 100 iterations with step-size $\eta = 0.1$. Figures 1,2,3 present the convergence results for different initializations $x_0 = 20, 5, -5$ respectively. In each figure, the first column shows the means and variances of the iterates along the trajectories; the center column shows the histograms of the last iterates and the target density (red curve); the last column shows the convergence of Wasserstein-2 distance along the trajectories. We also sample from the two-dimensional student-t distribution with center at the origin and $4$ degrees of freedom by running the proximal samplers with different oracles in parallel for 30 times. Each individual chain is run for 20 iterations with step-size $\eta = 0.1$ with the initialization at $x_0 = [5, 1]$. In Figure 4, we present the convergence results, the first column showing the means and variances of the first-coordinates along the trajectories, the center column showing the histograms of first-coordinate in the last iterates and the first-coordinate marginal density of the target distribution (red curve), and the last column showing the convergence of Wasserstein-2 distance along the trajectories.

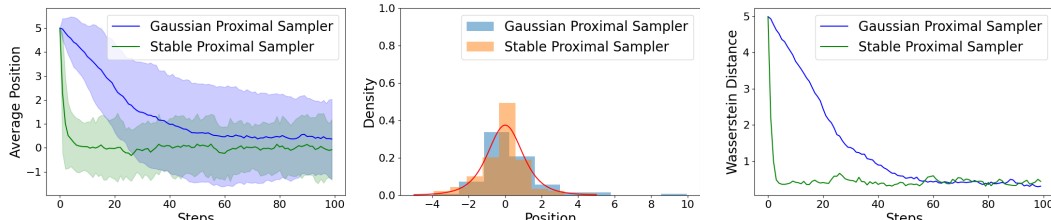

Figure 2: Comparison between Gaussian and Stable Proximal Sampler: target is chosen to be one-dimensional student-t with center $0$ and $4$ degrees of freedom; initialization is chosen $x_0 = 5$.

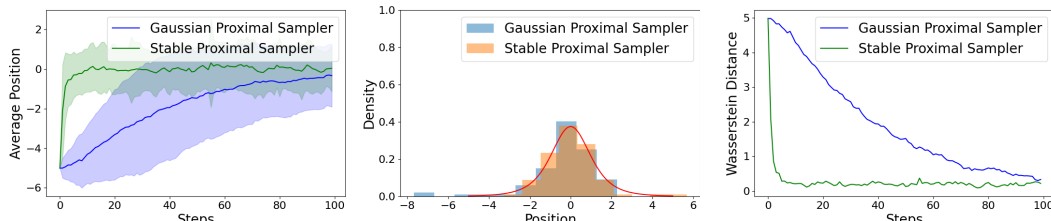

Figure 3: Comparison between Gaussian and Stable Proximal Sampler: target is chosen to be one-dimensional student-t with center $0$ and $4$ degrees of freedom; initialization is chosen $x_0 = -5$.

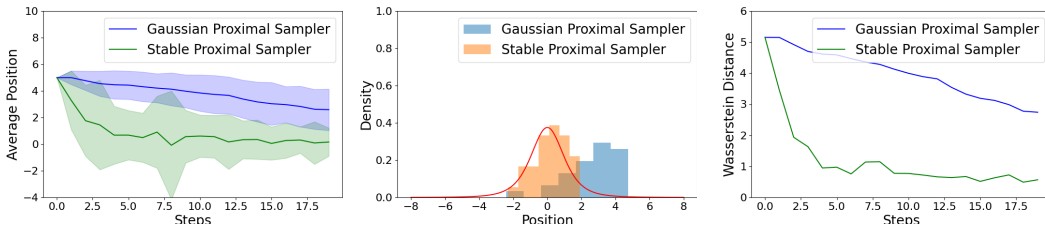

Figure 4: Comparison between Gaussian and Stable Proximal Sampler: target is chosen to be two-dimensional student-t with center $(0, 0)$ and $4$ degrees of freedom; initialization is chosen $x_0 = [5, 1]$.

