# OpenReview forum: "A Separation in Heavy-Tailed Sampling: Gaussian vs. Stable Oracles for Proximal Samplers"
_NeurIPS.cc/2024/Conference — NeurIPS 2024 poster_

### Official Review · Reviewer_kBax · 2024-07-04

**Soundness:** 3
**Presentation:** 4
**Contribution:** 3
**Rating:** 7
**Confidence:** 3

**Summary:**

The paper investigates the complexity of sampling from heavy-tailed distributions and presents a distinction between obtaining high-accuracy and low-accuracy guarantees. It analyzes two types of proximal samplers: those based on Gaussian oracles and those based on stable oracles. The main findings are that Gaussian oracle-based samplers can only achieve low-accuracy guarantees when sampling from heavy-tailed distributions, while stable oracle-based samplers can achieve high-accuracy guarantees. Additionally, the paper establishes lower bounds for samplers using the stable oracle, indicating that the presented upper bounds are optimal and cannot be fundamentally improved.

**Strengths:**

1. The problem is well-motivated and interesting.
2. Designed the algorithms and derived the upper bounds and lower bounds for different settings.
3. The authors also provided insightful discussion.
4. The authors provided solid theoretical proof for the results.

**Weaknesses:**

There is no experiment to verify the theoretical findings.

**Questions:**

1. Can you give an example in the real-world to motivate your problem?
2. Is it possible to run some experiments to verify your results?

**Limitations:**

There is no experiment.

---

> ### Author Rebuttal · Authors · 2024-08-07
>
> We sincerely thank the reviewer for their valuable advice and comments and greatly appreciate the positive evaluation.
>
> >Weakness: There is no experiment to verify the theoretical findings.
> >Question 2: Is it possible to run some experiments to verify your results?
>
> Following the reviewer's suggestion, we have added numerical experiments that compare the Gaussian proximal sampler and the stable proximal sampler with $\alpha=1$; see the uploaded pdf file as part of the rebuttal. In the first three experiments (corresponding to rows), we choose the target distribution to be the one-dimensional student-t distribution with 4 degrees of freedom and zero mean, and run the algorithms in parallel for 100 steps with step-size 0.1, to get 100 chains. We adopt different initializations, $x_0=20,5,-5$, and visualize the convergence via the average trajectories (with standard deviations), histogram of the last-step samples and the Wasserstein-2 distance decay, respectively. In the last experiment, we choose a 2-dimensional student-t distribution with 4 degrees of freedom and zero mean, and run the algorithms in parallel for 20 steps with step-size 0.1, to get 30 chains. We adopt the initialization, $x_0=[5,1]$, and use the same visualizations for the first-coordinate (marginals). The stable proximal sampler outperforms the Gaussian proximal sampler in all cases.
>
> In the revised version of our paper, we will include a section containing extensive detailed numerical studies demonstrating the performance of the algorithms.
>
> >Question 1: Can you give an example in the real-world to motivate your problem?
>
> There are several real-world applications of heavy-tailed sampling which arises in various domains such as Bayesian statistics [GJPS08, GLM18], machine learning [CDV09, BZ17, NSR19, SZTG20, DKTZ20], robust statistics [KN04, JR07, Kam18, YŁR22], multiple comparison procedures [GBH04, GB09], and study of geophysical systems [SP15, QM16, PBEM23].
>
> In particular the papers by [SP15, QM16, PBEM23] discuss real-world data analysis problems which crucially hinge on heavy-tailed sampling. We will be happy to elaborate this problem in the revision, subjected to space constraints.
>
> Furthermore, there has been recent works on heavy-tailed diffusion models; see e.g., [YPKL24],	arXiv:2407.18609 and [PSVM24]. We anticipate that our results in this work have implications for heavy-tailed diffusion models as well.
>
> Finally, please refer to the recent workshops at Neurips 2023 (titled "Heavy Tails in ML: Structure, Stability, Dynamics") and Issac Newton institute (titled "Heavy tails in machine learning") for further emerging applications of heavy-tailed sampling in real-world data science.
>
> [YPKL24]- Yoon, E. B., Park, K., Kim, S., & Lim, S. (2023). Score-based generative models with Lévy processes. Advances in Neural Information Processing Systems, 36, 40694-40707.
>
> [PSVM24] - Paquet, E., Soleymani, F., Viktor, H. L., & Michalowski, W. (2024). Annealed fractional Lévy–Itō diffusion models for protein generation. Computational and Structural Biotechnology Journal, 23, 1641-1653.

---

### Official Review · Reviewer_YAjf · 2024-07-10

**Soundness:** 3
**Presentation:** 3
**Contribution:** 3
**Rating:** 7
**Confidence:** 1

**Summary:**

This paper studies the problem of heavy-tailed sampling. First, the paper shows that while the gaussian proximal samplers are efficient for light-tailed targets, they are not accurate for heavy-tailed ones; the paper develops a lower bounds for the Gaussian proximal samplers, which reveals a fundamental challenge in heavy-tailed settings.

Then, the paper proceeds to develop a novel samplers based on restricted alpha-stable oracle; the insight is to replace the standard heat equation in gaussian oracle with a fractional heat flow. The paper proves that under suitable conditions the proposed sampler is efficient for heavy-tailed targets. Additionally, the paper proposes a practical implementation for a particular case of alpha=1.

**Strengths:**

- Novel theoretical analysis for the gaussian oracle sampler, which provides a new insight to developing sampling algorithms

- A novel methodology for heavy-tailed sampling

**Weaknesses:**

- The paper is purely theoretical and lacks experimental evaluation; it would be nice to at least have a toy illustration for the implementable algorithm 2+3 in the alpha=1 case.

- As the authors discussed in Sec5, the current paper does not present implementable algorithms for general alpha values in (0,2).

**Questions:**

- I wonder if the efficiency rejection sampling efficiency in Alg.3 has been taken into account of the sampler's theoretical complexity and practical complexity?

- Maybe I am missing this -- what is the impact of alpha?

**Limitations:**

Most of the limitations have been touched upon in sec 5. Otherwise see the weakness comments.

---

> ### Author Rebuttal · Authors · 2024-08-07
>
> We sincerely thank the reviewer for their valuable advice and comments and greatly appreciate the positive evaluation.
>
> >The paper is purely theoretical and lacks experimental evaluation; it would be nice to at least have a toy illustration for the implementable algorithm 2+3 in the $\alpha=1$ case.
>
> Following the reviewer's suggestion, we have added numerical experiments that compare the Gaussian proximal sampler and the stable proximal sampler with $\alpha=1$; see the uploaded pdf file as part of the rebuttal. In the first three experiments (corresponding to rows), we choose the target distribution to be the one-dimensional student-t distribution with 4 degrees of freedom and zero mean, and run the algorithms in parallel for 100 steps with step-size 0.1, to get 100 chains. We adopt different initializations, $x_0=20,5,-5$, and visualize the convergence via the average trajectories (with standard deviations), histogram of the last-step samples and the Wasserstein-2 distance decay, respectively. In the last experiment, we choose a 2-dimensional student-t distribution with 4 degrees of freedom and zero mean, and run the algorithms in parallel for 20 steps with step-size 0.1, to get 30 chains. We adopt the initialization, $x_0=[5,1]$, and use the same visualizations for the first-coordinate (marginals). The stable proximal sampler outperforms the Gaussian proximal sampler in all cases.
>
> In the revised version of our paper, we will include a section containing extensive detailed numerical studies demonstrating the performance of the algorithms.
>
> >As the authors discussed in Sec5, the current paper does not present implementable algorithms for general alpha values in (0,2).
>
> There is currently a difficulty to implement the R$\alpha$SO exactly for general $\alpha\in (0,2)$ due to the fact that there is no explicit representation for the $\alpha$-stable density for general $\alpha$. It is an interesting to investigate an exact/inexact implementation of the R$\alpha$SO for general $\alpha\in (0,2)$.
>
> >I wonder if the efficiency rejection sampling efficiency in Alg.3 has been taken into account of the sampler's theoretical complexity and practical complexity?
>
> Similar to the discussion of the Gaussian proximal sampler in [1], the iteration complexity results in our paper assumes the R$\alpha$SO, and the efficiency of Algorithm 3 is not included. In section 3, we provide Algorithm 3 as a practical implementation of R$\alpha$SO and its efficiency is discussed in Remark 3.
>
> >Maybe I am missing this -- what is the impact of $\alpha$?
>
> $\alpha$ is a parameter that appears in the Fractional Poincare inequality (FPI) and the stable process. As shown in Theorem 3, if the target distribution satisfies $\alpha$-FPI, the $\alpha$-stable proximal sampler convergens exponentially fast. The $\alpha$ in $\alpha$-FPI characterizes the tail-heaviness of the target distribution. If the target is extreme heavy-tail, we need to apply the stable proximal sampler with a small value of $\alpha\in (0,2)$ so that the target satisfies the $\alpha$-FPI. If we choose a large $\alpha\in (0,2)$ and the target doesn't satisfy $\alpha$-FPI, then the $\alpha$-stable proximal sampler only converges polynomially fast. Details are included in Section B.4 in the appendix.
>
> [1] Chen, Yongxin, et al. "Improved analysis for a proximal algorithm for sampling." Conference on Learning Theory. PMLR, 2022

---

> > ### Comment · Reviewer_YAjf · 2024-08-12
> >
> > I would like to thank the reviewers for their detailed reply.
> >
> > My concerns are mostly addressed. Also the empirical results seem promising. I will increase my rating to 7.

---

### Official Review · Reviewer_vTyM · 2024-07-12

**Soundness:** 2
**Presentation:** 2
**Contribution:** 3
**Rating:** 7
**Confidence:** 1

**Summary:**

The paper focus on studying the complexity of heavy-tailed sampling and present a separation result in terms of obtaining high-accuracy versus low-accuracy guarantees. Their results are presented for proximal samplers that are based on Gaussian versus stable oracles. Authors show that proximal samplers based on the Gaussian oracle have a fundamental barrier in that they necessarily achieve only low-accuracy guarantees when sampling from a class of heavy-tailed targets. In contrast, proximal samplers based on the stable oracle exhibit high-accuracy guarantees, thereby overcoming the aforementioned limitation. They also prove lower bounds for samplers under the stable oracle and show that our upper bounds cannot be fundamentally improved.

**Strengths:**

Although I am not an expert in this field, I find this work quite interesting. The authors provide new material and support their statements with proofs.

**Weaknesses:**

The paper is not tested in any way on a numerical experiment. I am convinced that a paper presented at this type of conference should be both motivated by a real-world application and tested numerically, e.g., on a near-real-world formulation of the problem.

**After a rebuttal process**, the authors agreed with this weakness and promised to add the experiments to the final version of the paper.

**Questions:**

N/A

---

> ### Author Rebuttal · Authors · 2024-08-07
>
> We sincerely thank the reviewer for their valuable advice and comments and greatly appreciate the positive evaluation.
>
> >The paper is not tested in any way on a numerical experiment. I am convinced that a paper presented at this type of conference should be both motivated by a real-world application and tested numerically, e.g., on a near-real-world formulation of the problem.
>
> Following the reviewer's suggestion, we have added numerical experiments that compare the Gaussian proximal sampler and the stable proximal sampler with $\alpha=1$; see the uploaded pdf file as part of the rebuttal. In the first three experiments (corresponding to rows), we choose the target distribution to be the one-dimensional student-t distribution with 4 degrees of freedom and zero mean, and run the algorithms in parallel for 100 steps with step-size 0.1, to get 100 chains. We adopt different initializations, $x_0=20,5,-5$, and visualize the convergence via the average trajectories (with standard deviations), histogram of the last-step samples and the Wasserstein-2 distance decay, respectively. In the last experiment, we choose a 2-dimensional student-t distribution with 4 degrees of freedom and zero mean, and run the algorithms in parallel for 20 steps with step-size 0.1, to get 30 chains. We adopt the initialization, $x_0=[5,1]$, and use the same visualizations for the first-coordinate (marginals). The stable proximal sampler outperforms the Gaussian proximal sampler in all cases.
>
> In the revised version of our paper, we will include a section containing extensive detailed numerical studies demonstrating the performance of the algorithms.

---

> > ### Comment · Reviewer_vTyM · 2024-08-08
> >
> > Dear Authors,
> >
> > Thank you for the reply. The responses are satisfactory. I am raising my score by +1.

---

### Official Review · Reviewer_MHrz · 2024-07-13

**Soundness:** 3
**Presentation:** 3
**Contribution:** 3
**Rating:** 8
**Confidence:** 3

**Summary:**

The authors provide a lower bound for sampling from heavy tailed distributions under the Gaussian oracle of order $O(\textup{poly}(1/\varepsilon))$. They then propose an alternative proximal sampling algorithm using the $\alpha$-stable oracle that achieves a convergence rate of $O(\log(1/\varepsilon))$ for heavy-tailed distributions satisfying a fractional Poincare inequality. They then provide a practical implementation of the stable proximal sampler, and lower bounds on its convergence rate.

**Strengths:**

- This work presents a very nice combination of results showing a separation in the performance of stable and Gaussian proximal samplers. The combination of lower and upper bounds separating the two methods makes the work a particularly interesting contribution.

- The addition of a practical implementation of the stable proximal sampler is nice to have, demonstrating that it is viable in practice.

- The work is generally clearly presented and the authors are clear about their contributions.

- Overall, I consider this to be a very sound piece of theoretical work.

**Weaknesses:**

I have no major concerns about this paper. The presentation is somewhat dense in places, though this is mostly just a consequence of it being a very technical paper and not a flaw as such. If the authors want to make the claim that practicioners should use the stable proximal sampler in applied settings, then they may want to provide empirical evidence of its performance compared to the Gaussian proximal sampler. However, I understand that this is not the main purpose of this theoretical paper.

**Questions:**

I have no clarifications to request.

**Limitations:**

The authors provide an adequate discussion of the limitations of their methods in the final section, and I foresee no additional negative impacts of their work.

---

> ### Author Rebuttal · Authors · 2024-08-07
>
> We sincerely thank the reviewer for their valuable advice and comments and greatly appreciate the positive evaluation.
>
> >I have no major concerns about this paper. The presentation is somewhat dense in places, though this is mostly just a consequence of it being a very technical paper and not a flaw as such. If the authors want to make the claim that practicioners should use the stable proximal sampler in applied settings, then they may want to provide empirical evidence of its performance compared to the Gaussian proximal sampler. However, I understand that this is not the main purpose of this theoretical paper.
>
> Following the reviewer's suggestion, we have added numerical experiments that compare the Gaussian proximal sampler and the stable proximal sampler with $\alpha=1$; see the uploaded pdf file as part of the rebuttal. In the first three experiments (corresponding to rows), we choose the target distribution to be the one-dimensional student-t distribution with 4 degrees of freedom and zero mean, and run the algorithms in parallel for 100 steps with step-size 0.1, to get 100 chains. We adopt different initializations, $x_0=20,5,-5$, and visualize the convergence via the average trajectories (with standard deviations), histogram of the last-step samples and the Wasserstein-2 distance decay, respectively. In the last experiment, we choose a 2-dimensional student-t distribution with 4 degrees of freedom and zero mean, and run the algorithms in parallel for 20 steps with step-size 0.1, to get 30 chains. We adopt the initialization, $x_0=[5,1]$, and use the same visualizations for the first-coordinate (marginals). The stable proximal sampler outperforms the Gaussian proximal sampler in all cases.
>
> In the revised version of our paper, we will include a section containing extensive detailed numerical studies demonstrating the performance of the algorithms.

---

> > ### Comment · Reviewer_MHrz · 2024-08-07
> >
> > Thank you for your comment. I appreciate the engagement with the comments and additional experimental results provided.

---

### Official Review · Reviewer_nmvc · 2024-07-13

**Soundness:** 3
**Presentation:** 3
**Contribution:** 3
**Rating:** 6
**Confidence:** 2

**Summary:**

This paper studies the complexity of sampling heavy-tailed distributions. It provides lower bounds on the complexity of Gaussian-based samplers for a class of heavy-tailed targets. Then, the paper constructs proximal samplers based on stable oracles, which improve the sampling complexity.

**Strengths:**

* This paper is well-written. The background of sampling and the research problems regarding sampling complexity are clearly introduced. The contributions of the lower bound on Gaussian-based samplers for heavy-tailed targets and the improved complexity using stable oracles are clearly presented.
* The paper is technically sound. The definitions and assumptions are discussed clearly, and the theoretical results are supported by proof sketches.

**Weaknesses:**

The contribution of the paper could be improved with empirical experiments to evaluate the sampling algorithms and their complexity.

**Questions:**

* Is there any intuition that a Gaussian-based sampler has lower accuracy for heavy-tailed targets than for non-heavy-tailed targets?
* How would a Gaussian-based sampler compare with a stable oracle for not heavy-tailed targets?

---

> ### Author Rebuttal · Authors · 2024-08-07
>
> We sincerely thank the reviewer for their valuable advice and comments and greatly appreciate the positive evaluation.
>
> >The contribution of the paper could be improved with empirical experiments to evaluate the sampling algorithms and their complexity.
>
> Following the reviewer's suggestion, we have added numerical experiments that compare the Gaussian proximal sampler and the stable proximal sampler with $\alpha=1$; see the uploaded pdf file as part of the rebuttal. In the first three experiments (corresponding to rows), we choose the target distribution to be the one-dimensional student-t distribution with 4 degrees of freedom and zero mean, and run the algorithms in parallel for 100 steps with step-size 0.1, to get 100 chains. We adopt different initializations, $x_0=20,5,-5$, and visualize the convergence via the average trajectories (with standard deviations), histogram of the last-step samples and the Wasserstein-2 distance decay, respectively. In the last experiment, we choose a 2-dimensional student-t distribution with 4 degrees of freedom and zero mean, and run the algorithms in parallel for 20 steps with step-size 0.1, to get 30 chains. We adopt the initialization, $x_0=[5,1]$, and use the same visualizations for the first-coordinate (marginals). The stable proximal sampler outperforms the Gaussian proximal sampler in all cases.
>
> In the revised version of our paper, we will include a section containing extensive detailed numerical studies demonstrating the performance of the algorithms.
>
> >Is there any intuition that a Gaussian-based sampler has lower accuracy for heavy-tailed targets than for non-heavy-tailed targets?
>
> For the Gaussian-based sampler, the repeated sampling steps $y_k\sim \pi^{Y|X}(\cdot|x_k)=\mathcal{N}(x_k,\eta I_d)$ are crucial to ensure rapid convergence. These are essentially heat flow simulations, that take advantage of the geometric properties of the logconcave (non-heavy-tailed) target distributions, providing an exponential decay in KL-divergence.
>
> When the target distribution is heavy-tailed, the standard heat flow mixes slowly. Intuitively, if our initial condition is a light-tailed random variable, it would take many steps for the Gaussian proximal sampler to match the moments of the heavy-tailed target, because in every iteration we only add Gaussian noise with a small variance.
>
> >How would a Gaussian-based sampler compare with a stable oracle for not heavy-tailed targets?
>
> Gaussian proximal sampler performs well when the target satisfies a Poincare or alog-Sobolev inequality [1]. When the RGO step is implemented exactly, the algorithm converges exponentially fast. Since these light-tailed distributions also satisfy the Fractional Poincare inequality (FPI), our Theorem 3 suggests the stable proximal sampler also converges exponentially fast. The difference lies in the rates of exponential convergence. Intuitively, stable proximal sampler may have a smaller convergence rate compared to Gaussian proximal sampler. For example, if the target is a Gaussian which has a concentrated region that most samples would lie in, since the stable proximal sampler adds heavy-tail randomnesses in each iteration, it may move a sample out of the concentrated region more easily. Therefore, it takes longer for the stable proximal sampler to move most of the particles inside the concentrated region. Analytically, the convergence rates depend on the Poincare constant and the FPI constant in the Gaussian and stable case, respectively
>
> [1] Chen, Yongxin, et al. "Improved analysis for a proximal algorithm for sampling." Conference on Learning Theory. PMLR, 2022.

---

### Author Rebuttal · Authors · 2024-08-07

We sincerely thank the reviewer for their valuable advice and comments and greatly appreciate the positive evaluation. Results from the added experiments are included in the pdf file.

---

### Decision · Program_Chairs · 2024-09-25

**Decision:**

Accept (poster)

**Comment:**

The paper presents a separation between Gaussian and stable-based proximal samplers in terms of low vs. high-accuracy guarantees. All reviewers recommended acceptance after discussion. The main concern was lack of experiments, which the authors presented during discussion. I recommend acceptance and remind the authors to add the experiments to the paper.